# Hierarchical Reinforcement Learning With Timed Subgoals

**Nico Gürtler**    **Dieter Büchler**    **Georg Martius**

Max Planck Institute for Intelligent Systems
Tübingen, Germany
{nguertler, dbuechler, gmartius}@tue.mpg.de

## Abstract

Hierarchical reinforcement learning (HRL) holds great potential for sample-efficient learning on challenging long-horizon tasks. In particular, letting a higher level assign subgoals to a lower level has been shown to enable fast learning on difficult problems. However, such subgoal-based methods have been designed with static reinforcement learning environments in mind and consequently struggle with dynamic elements beyond the immediate control of the agent even though they are ubiquitous in real-world problems. In this paper, we introduce Hierarchical reinforcement learning with Timed Subgoals (HiTS), an HRL algorithm that enables the agent to adapt its timing to a dynamic environment by not only specifying what goal state is to be reached but also *when*. We discuss how communicating with a lower level in terms of such *timed subgoals* results in a more stable learning problem for the higher level. Our experiments on a range of standard benchmarks and three new challenging dynamic reinforcement learning environments show that our method is capable of sample-efficient learning where an existing state-of-the-art subgoal-based HRL method fails to learn stable solutions.[1]

## 1  Introduction

Hierarchical reinforcement learning (HRL) has recently begun to live up to its promise of sample-efficient learning on difficult long-horizon tasks. The idea behind HRL is to break down a complex problem into a hierarchy of more tractable subtasks. A particularly successful approach to defining such a hierarchy is to let a high-level policy choose a subgoal which a low-level policy is then tasked with achieving [8]. Due to the resulting temporal abstraction such *subgoal-based* HRL methods have been shown to be able to learn demanding tasks with unprecedented efficiency [31, 23, 19].

In order to realize the full potential of HRL it is necessary to design algorithms that enable concurrent learning on all levels of the hierarchy. However, the changing behavior of the lower level during training introduces a major difficulty. From the perspective of the higher level, the reinforcement learning environment and the policy on the lower level constitute an effective environment which determines what consequences its actions will have. During training, the learning progress on the lower level renders this effective environment non-stationary. If this issue is not addressed, the higher level will usually not start to learn efficiently before the lower level is fully converged. This situation is similar to a manager and a worker trying to solve a task together while the meaning of the vocabulary they use for communication is continuously changing. Clearly, a stable solution can then only be found once the worker reacts reliably to instructions. Hence, to enable true concurrent learning, all

---

[1]Videos and code, including our algorithm and the proposed dynamic environments, can be found at https://github.com/martius-lab/HiTS.

levels in a hierarchy should see transitions that look like they were generated by interacting with a stationary effective environment.

Existing algorithms partially mask the non-stationarity of the effective environment by replacing the subgoal chosen by the higher level appropriately in hindsight. Combined with the subtask of achieving or progressing towards the assigned subgoal as fast as possible, this approach was shown to enable fast learning on a range of challenging sparse-reward, long-horizon tasks [23, 19]. What these methods do not take into account, however, is that, if adaptive temporal abstraction is used, the higher level in the hierarchy is effectively interacting with a semi-Markov decision process (SMDP), i.e., transition times vary. If the objective of the lower level is to reach a subgoal as fast as possible, then the amount of time that elapses until it reaches a given subgoal and returns control to the higher level will decrease during training. Hence, the distribution of the transition times the higher level sees will shift to lower values which introduces an additional source of non-stationarity. When trying to quickly traverse a static environment, such as a maze, this shift is in line with the overall task and will contribute to the learning progress.

Yet, as soon as dynamic elements that are beyond the immediate control of the agent are present, the situation changes radically. Consider, for example, the task of returning a tennis ball to a specified point on the ground by hitting it with a racket. This clearly requires the agent to time its actions so as to intercept the ball trajectory with the racket while it has the right orientation and velocity. Even if the higher level found a sequence of subgoals (specifying the state of the racket) that brought about the right timing, this solution would stop working as soon as the lower level learns to reach them faster. This would require the higher level to choose a different and possibly longer sequence of subgoals, a process that would continue until the lower level was fully converged. Hence, exposing the higher level to a non-stationary distribution of transition times will lead to training instability and slow learning. As it is the rule rather than the exception for real-world environments to contain dynamic elements beyond the immediate control of the agent – think of humans collaborating with a robot or an autonomous car navigating traffic – this problem can be expected to hinder the application of HRL to real-world tasks.

In order to solve the non-stationarity issue in dynamic environments, we propose to let the higher level choose not only what subgoal is to be reached but also *when*. By emitting such *timed subgoals*, consisting of a desired subgoal and a time interval that is supposed to elapse before it is reached, the higher level has explicit control over the transition times of the SMDP it interacts with. This completely removes the non-stationarity of transition times and allows for stable concurrent learning in dynamic environments when combined with a technique for hiding the non-stationarity of transitions in the subgoal space. It furthermore gives the higher level explicit control over the degree of temporal abstraction, giving it more direct access to the trade off between a small effective problem horizon and exercising tight control over the agent.

The main contribution of this work is to distill these insights into the formulation of a sample-efficient HRL algorithm based on timed subgoals (Section 3) which we term Hierarchical reinforcement learning with Timed Subgoals (HiTS). We demonstrate that HiTS is competitive on four standard benchmark tasks and propose three novel tasks that exemplify the challenges introduced by dynamic elements in the environment. While HiTS succeeds in solving them, prior methods fail to learn a stable solution (Section 4). In a theoretical analysis, we show that the use of timed subgoals in combination with hindsight action relabeling [19] removes the non-stationarity of the SMDP generating the data the higher level is trained on (Section 3).

## 2 Background

### 2.1 Reinforcement learning and subgoal-based hierarchical reinforcement learning

We assume the reinforcement learning problem is formulated as a Markov decision process (MDP) defined by a tuple $(\mathcal{S}, \mathcal{A}, p, r, \rho_0, \gamma)$ consisting of state space $\mathcal{S}$, action space $\mathcal{A}$, transition probabilities $p(s' \mid s, a)$, reward function $r(s, a)$, initial state distribution $\rho_0(s_0)$ and discount factor $\gamma \in [0, 1)$. At each time step, the agent is provided with a state $s_t \in \mathcal{S}$ and samples an action $a_t \in \mathcal{A}$ from the distribution $\pi(\cdot \mid s_t)$ defined by its policy $\pi$. The environment then provides feedback in the form of a reward $r(s_t, a_t) \in \mathbb{R}$ and transitions to the next state $s_{t+1} \in \mathcal{S}$. The objective is to find an optimal

policy $\pi^*$ that maximizes the expected discounted return

$$J(\pi) = \mathbb{E}_{\pi, s_0 \sim \rho_0} \left[ \sum_{t=0}^{\infty} \gamma^t r(s_t, a_t) \right] .$$ (1)

In this section, we consider the subgoal-based HRL approach to solving this problem where a lower level is tasked with achieving a subgoal provided by a higher level [8, 23, 19]. The lower level (index 0) acts directly on the environment by outputting a primitive action in $\mathcal{A}$, whereas the higher level (index 1) proposes subgoals for level 0. Hence, the action of level 1 is the subgoal for level 0, i.e. $g^0 := a_t^1$. Since level 0 performs a variable number of steps $\tau$ in the environment before "finishing" its subtask, level 1 effectively interacts with a semi-Markov decision process (SMDP) [16, 17]. Its transition probabilities $p(s_{t+\tau}, \tau \mid s_t, a_t^1)$ determine the distribution not only of the next state $s_{t+\tau}$ level 1 observes but also of the time $\tau$ that elapses beforehand.

Existing subgoal-based algorithms reward the lower level either for being close to the subgoal [23] or for progressing in a given direction in subgoal space [31] or penalize every action that does not lead to the immediate achievement of the subgoal [19]. In this section, we consider the latter case of a shortest path objective with a reward

$$r^0(s', g^0) = \begin{cases} 0 & \text{if } \phi(s', g^0) = 1, \\ -1 & \text{otherwise} \end{cases}$$ (2)

for transitioning to a new state $s'$. The function $\phi(s', g^0)$ evaluates to 1 only if $s'$ is sufficiently close to the subgoal $g^0$ in some metric (e.g. Euclidean distance). The higher level is queried for a new subgoal either if the lower level achieved a state sufficiently close to the subgoal or if the lower level has used up a fixed budget of actions.

## 2.2 Non-stationarity of the induced SMDP and hindsight action relabeling

The higher level of the hierarchy interacts with an SMDP that is induced by the environment dynamics and the behaviour of the lower level. Changes to the low-level policy during training render this SMDP non-stationary which prevents the higher level from learning a stable solution. Applying the recently proposed hindsight action relabeling technique can alleviate this problem for static environments [19]. We show that for dynamic environments the non-stationarity reappears.

For reasons of sample-efficiency, data should be reused after a policy update instead of being discarded. We therefore consider training with an off-policy algorithm that uses a replay buffer. On level 1 transitions of the form $(s, a^1, r^1, s')$ are being stored, containing the state $s$, the action $a^1$, the reward $r^1$ and the state $s'$ at the end of the subtask execution. For now, we assume that the action space of level 1 (which is equal to the goal space of level 0) is the full state space.

Due to the changing low-level policy $\pi^0$, the distribution of the next state $s'$ and the reward $r^1$ given a fixed state action pair $(s, a^1)$ will change in the course of training. Hence, the SMDP level 1 interacts with is non-stationary and learning a stable high-level policy becomes feasible only after the lower level is fully converged. *Hindsight action relabeling*, as proposed in [19], addresses this problem by "pretending" that the low-level policy is already optimal, i.e., is able to reach assigned subgoals. This is achieved by replacing the high-level action (corresponding to the assigned subgoal) with the state the lower level actually achieved before storing transitions in the replay buffer. If the reward depends on the action, it has to be adapted as well:

$$(s, a^1, r^1, s') \implies (s, \hat{a}^1 := s', \hat{r}^1 := r^1(s, \hat{a}^1), s') \qquad \text{(hindsight action relabeling).}$$ (3)

In Fig. 1a we illustrate this relabeling procedure.

**Proposition 1** *Applying hindsight action relabeling (Eq. 3) at level 1 generates transitions that follow a stationary state transition and reward distribution, i.e. $p_t \left( s, \hat{a}^1, \hat{r}^1, s' \right) = p_t \left( s, \hat{a}^1 \right) p \left( s', \hat{r}^1 \mid s, \hat{a}^1 \right)$ where $p \left( s', \hat{r}^1 \mid s, \hat{a}^1 \right)$ is time-independent provided that the subgoal space is equal to the full state space.*

The proof is given in Suppl. A. The remaining caveat is that the higher level should learn to restrict itself to subgoals which are feasible for the lower level. By conducting *testing transitions* [19], which

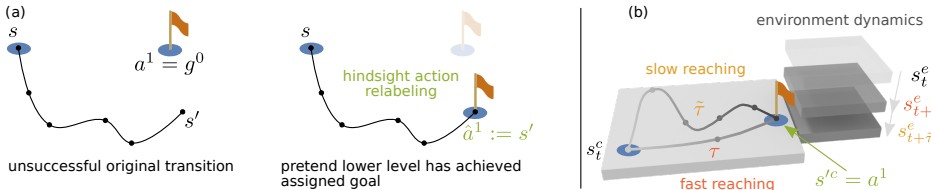

Figure 1: Relabeling results in a stationary state-transition distribution in static environments, but not in dynamic ones. (a) Action relabeling replaces the high-level action $a^1$ with the achieved state $s'$ as if the lower level was already optimal. (b) In dynamic environments with a state that factorizes into a directly controllable part $s^c$ and a remaining part $s^e$ with dynamics, action relabeling does not prevent different environment outcomes for slow and fast low-level policies if the subgoal determines only $s^c$. In this example, the agent tries to reach a moving platform.

are exempt from hindsight action relabeling and penalize infeasible subgoals, an appropriate incentive for the higher level can be introduced. Note that this reintroduces a non-stationarity to the data in the replay buffer. For the remaining part of this section we do not consider testing transitions as they are not directly related to the issue we are concerned with.

**Dynamic environments.** We now consider environments with dynamic elements which are not directly controllable by the agent, for instance a flying ball or a moving platform. In this case, choosing the full state space as the subgoal space is problematic as it would include parts of the state the agent has no control over. Consequently, the higher level would be forced to learn the full dynamics of the environment in order to be able to propose feasible subgoals. In such a setting it is therefore more appropriate to restrict the subgoal space to the directly controllable part of the state. Formally, we assume the state space to factorize as $\mathcal{S} = \mathcal{S}^c \times \mathcal{S}^e$ into a directly controllable part $\mathcal{S}^c$, e.g. the agent, and a part $\mathcal{S}^e$ corresponding to the rest of the environment [18, 15, 11]. For simplicity, we assume that $s^c$ does not influence $s^e$ in this section, i.e., $p\left(s'^e \mid s^e\right) = p(s'^e \mid s^e, s^c, a^0)$. A typical choice for a subgoal space would then be $\mathcal{S}^c$, i.e. $\mathcal{G}^0 = \mathcal{S}^c$, as it allows the lower level to focus on controlling the agent alone and allows for transfer to related tasks. However, due to the restriction of the subgoal space, Prop. 1 does not apply anymore.

In particular, hindsight action relabeling does not result in a stationary state-transition distribution anymore as illustrated in Fig. 1 (b). Even though relabeling the action of level 1 according to $\hat{a}^1 = s'^c$ hides the influence of the lower level on the achieved goal, the state of the environment $s'^e$ varies according to how long the lower level was in control. This transition time $\tau$ will shrink during the course of training as the lower level tries to optimize a shortest path objective. As a consequence, the same relabeled state-action pair $(s_t, \hat{a}^1 = s'^c)$ comes with different environmental outcomes $s^e_{t+\tau}$ and $s^e_{t+\tilde{\tau}}$ depending on the learning progress of the lower level. This makes it impossible for the agent to adapt its behavior to the environment before the lower level is fully converged. As a consequence, existing subgoal-based HRL methods with adaptive temporal abstraction struggle with dynamic environments.

## 3   Hierarchical reinforcement learning with timed subgoals

An observation that points to a way out of the non-stationarity dilemma in dynamic environments is that the readily controllable part often only sparsely influences the rest of the environment. In the example from the last section, the dynamics of the moving platform in Fig. 1 (b) is not affected by the agent at all. As a result, knowing the elapsed time alone completely determines the distribution of $s^e$, no matter how complex its dynamics may be. We therefore propose to let time stand in for $s^e$ and to condition the lower level on a *timed subgoal* $(g^0, \Delta t^0)$ where the desired time until achievement $\Delta t^0 \in \mathbb{N}_{>0}$ determines *when* the lower level is supposed to reach the subgoal state $g^0 \in S^c$.

Hence, after being assigned a timed subgoal at time $t$, the lower level stays in control until $t + \Delta t^0$. This definition of the hierarchy fixes the transition time the higher level sees to $\tau = \Delta t^0$. Combined with the assumption that $s^e$ evolves independently of $s^c$ and when replacing the action $a^1$ via hindsight action relabeling as discussed in Section 2.2, this completely removes the non-stationarity of the SMDP the level is interacting with.

**Proposition 2** *If the not directly controllable part of the environment evolves completely independently of the controllable part, i.e., if $p\left(s'^e \mid s^e\right) = p(s'^e \mid s^e, s^c, a^0)$, and if hindsight action relabeling is used, the transitions in the replay buffer of a higher level assigning timed subgoals to a lower level are consistent with a completely stationary SMDP. Thus, they follow a distribution $p_t\left(s, \hat{a}^1, \tau, s', \hat{r}^1\right) = p_t\left(s, \hat{a}^1\right) p\left(s', \tau, \hat{r}^1 \mid s, \hat{a}^1\right)$ where $p\left(s', \tau, \hat{r}^1 \mid s, \hat{a}^1\right)$ is time-independent.*

The proof is given in Suppl. A. Intuitively, the non-stationarity is removed because hindsight action relabeling hides the influence of the low-level policy on the state $s'^c$ the agent transitions to while fixing the time interval $\Delta t$ during which the lower level is in control makes sure that the low-level policy does not affect $s'^e$.

Of course, the assumption of an environment which is not at all influenced by the agent is restrictive. However, the interactions between agent and environment are often sparse in time (e.g. in the case of a racket hitting a ball). The higher level can then learn to identify the situations in which it has to exercise tight control over the agent because it is about to influence the environment and align its choice of desired subgoal achievement times to them. It then has full control over the interactions between agent and environment and the episode is divided into a sequence of time intervals during which the assumption of independent dynamics and consequently also Proposition 2 hold (see Fig. 2 (a)). From this point on, the transitions added to the replay buffer of the higher level will be consistent with a stationary SMDP again.

### 3.1   The HiTS algorithm

In this section, we present *Hierarchical reinforcement learning with Timed Subgoals* (HiTS), an HRL algorithm for sample-efficient learning on dynamic environments. We consider a two-level hierarchy in which the higher level assigns timed subgoals to the lower level as illustrated in Fig. 2 (b). For implementation details we refer to Suppl. B.

#### 3.1.1   The higher level

The high-level policy $\pi^1\left(\cdot \mid s, g\right)$ is conditioned on the state and optionally on the episode goal and outputs a timed subgoal $a^1 = \left(g^0, \Delta t_t^0\right) \in \mathcal{G}^0 \times \mathbb{N}_{>0}$. The lower level then pursues this timed subgoal for $\Delta t_t^0$ time steps after which the higher level is queried again. The objective of the higher level is to maximize the expected environment return while maintaining temporal abstraction. It therefore receives the cumulative environment reward plus a penalty $-c < 0$ for emitting a subgoal,

$$r^1\left(s_{t:t+\Delta t_t^0-1}, a_{t:t+\Delta t_t^0-1}\right) = \sum_{n=0}^{\Delta t_t^0-1} \gamma^n r\left(s_{t+n}, a_{t+n}\right) - c \,. \tag{4}$$

Note that for Proposition 2 to hold, the reward $r^1$ should neither depend on the atomic actions $a_t$ nor on the state $s^c$ except for when a timed subgoal is achieved.

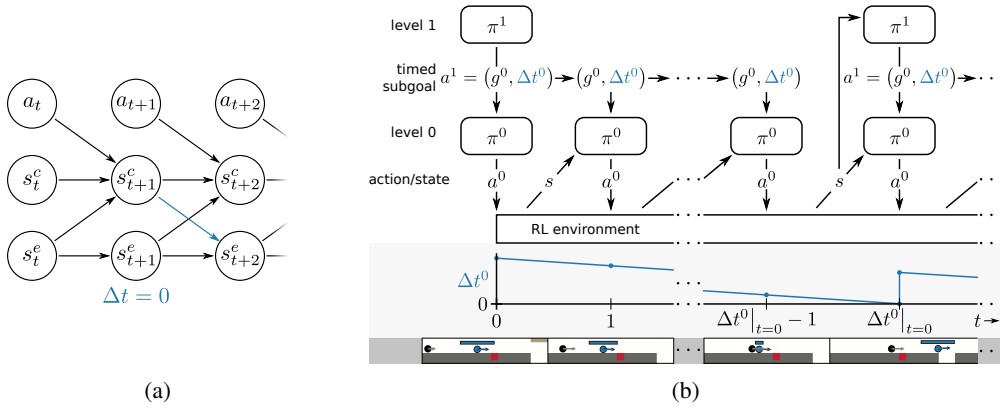

           (a)                                              (b)

Figure 2: (a) The influence of $s^c$ on $s^e$ is assumed to be sparse in time so that the higher level can align timed subgoals to it. (b) Execution of the HiTS algorithm over time.

### 3.1.2 The lower level

The subtask assigned to the lower level is to achieve the timed subgoal $a^1 = (g^0, \Delta t_t^0)$. Note that the desired achievement time $\Delta t_t^0$ is given relative to the current time $t$, i.e., the goal state $g^0$ is to be achieved at the time $t + \Delta t_t^0$. It is therefore decremented in each time step, $\Delta t_{t+1}^0 = \Delta t_t^0 - 1$, before being passed to the policy. We chose this representation as the dynamics of an agent usually do not have an explicit time dependence and consequently time intervals rather than absolute times are relevant to control. In other words, a policy conditioned on this representation of a timed subgoal in combination with a time-invariant environment is automatically time-invariant as well.

A state is translated into an achieved subgoal by a map $f^{\mathcal{G}} : \mathcal{S} \to \mathcal{G}^0$. In the setting of a factorized state space $\mathcal{S} = \mathcal{S}^c \times \mathcal{S}^e$ the mapping may simply project onto the directly controllable part $\mathcal{S}^c$. The policy is conditioned on the observation $o_t^0$ (e.g. $s^c$) and the desired timed subgoal and defines a density $\pi^0(\cdot \mid o_t^0, g^0, \Delta t_t^0)$ in action space. During execution of the hierarchy, an action $a_t^0 \in \mathcal{A}$ is sampled from this distribution and passed on to the environment. If the desired time until achievement has run out, i.e., if $\Delta t_{t+1}^0 = 0$, the higher level is queried for a new subgoal (see Fig. 2 (b)).

During training, the lower level receives a non-zero reward only if it achieves a timed subgoal, i.e., if the achieved subgoal is sufficiently close to the desired subgoal at the desired time of achievement. Hence, the reward in a transition to a new state $s_{t+1}$ reads

$$r_t^0 = \begin{cases} 1 & \text{if } \phi(f^{\mathcal{G}}(s_{t+1}), g^0) = 1 \text{ and } \Delta t_{t+1}^0 = 0, \\ 0 & \text{otherwise,} \end{cases} \qquad (5)$$

where $\phi : \mathcal{G}^2 \to [0, 1]$ specifies whether the achieved and the desired subgoal are sufficiently close to each other. Note that when learning the value function of the lower level we consider $s_{t+1}$ as a terminal state when $\Delta t_{t+1}^0 = 0$ is reached, i.e., we do not bootstrap and set $\gamma^0 = 0$.

### 3.1.3 Off-policy training and use of hindsight

In order to realize the potential of HRL for sample-efficient learning, past experience should be reused instead of being discarded after each policy update. Hence, we use the off-policy reinforcement learning algorithm Soft Actor-Critic (SAC) [13] to train the policies on both levels. We address the non-stationarity issue by applying hindsight action relabeling to transitions before storing them in the replay buffer of the higher level,

$$(s_t, a_t^1 = (g_t^0, \Delta t_t^0), r_t, s_{t+\Delta t_t^0}) \implies (s_t, \hat{a}_t^1 := (\hat{g}_t^0, \Delta t_t^0), r^1(s_t, \hat{a}_t^1), s_{t+\Delta t_t^0}) , \qquad (6)$$

where $\hat{g}_t^0 = f^{\mathcal{G}}(s_{t+\Delta t_t^0})$ denotes the subgoal the lower level achieved. Note that the desired time until achievement $\Delta t_t^0$ does not have to be replaced as it is fixed by the higher level by definition. Under the assumption of a controllable part of the state which does not influence the rest of the environment, Proposition 2 ensures that the relabeled transitions in the replay buffer are consistent with a completely stationary SMDP.

In practice hindsight goal relabeling according to Eq. 6 is only applied if the desired subgoal was not achieved, i.e., if $\phi(\hat{g}^0, g^0) = 0$. This gives the higher level an idea of how precise the lower level is in achieving assigned subgoals while still hiding its failures. Furthermore, while pursuing a fixed percentage of timed subgoals, the lower level deterministically outputs the mean of its action distribution. If it still fails to achieve the desired goal, the higher level is penalized. Such testing transitions give the higher level an idea of the current capabilities of the low-level policy and therefore ensure that it assigns feasible subgoals [19]. While these two exceptions from hindsight action relabeling reintroduce a non-stationarity into the replay buffer of the higher level, they are necessary to make it respect the limitations of the lower level.

As the reward in Eq. 5 is sparse, we use *Hindsight Experience Replay* (HER) [2] to generalize from achieved to desired timed subgoals on the lower level. The resulting hindsight goal transitions are based on hindsight action transitions so as to conserve their stationarity property. When choosing an achieved subgoal at $n$ time steps in the future as hindsight goal, the low-level transition is relabeled as

$$(s_t, a_t^0, r_t, s_{t+1}, g_t^0, \Delta t_t^0) \implies (s_t, a_t^0, r_t, s_{t+1}, \hat{g}_t^0 := f^{\mathcal{G}}(s_{t+n}), \widehat{\Delta t}_t^0 := n) . \qquad (7)$$

To increase sample-efficiency, we also apply conventional HER on the higher level for goal-based environments.

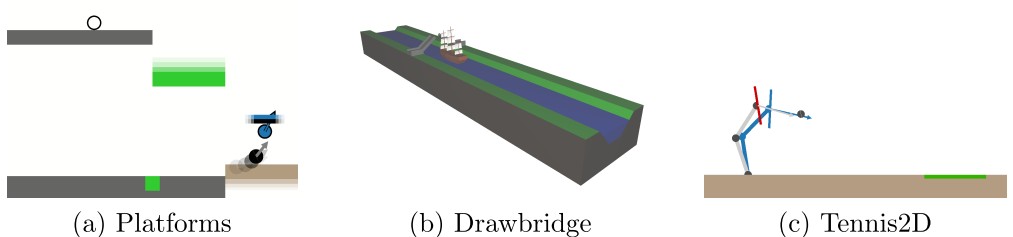

|                 |                   |                 |
| :-------------: | :---------------: | :-------------: |
| (a) Platforms   | (b) Drawbridge    | (c) Tennis2D    |

Figure 3: Dynamic environments. (a) To reach the goal, the agent has to trigger the movement of a platform at the right time. (b) In order not to collide with the drawbridge before it opens, the agent has to time the unfurling of its sails correctly. (c) A tennis ball is to be returned by a robot arm to a varying goal region.

## 4    Experiments

The goal of our experimental evaluation is to compare HiTS with prior subgoal-based methods in terms of sample efficiency and stability of learning. We evaluate on four standard benchmarks as well as three new reinforcement learning environments[2] that exemplify the challenges introduced by dynamic elements which are beyond the immediate control of the agent (see Fig. 3). The three proposed environments require the agent to find the shortest path (in terms of time steps) to the goal, i.e., the reward is $-1$ in every time step unless the goal was achieved (in which case it is $0$). If the agent did not reach the goal after a predefined number of time steps, the episode is terminated as well.

**Platforms.**  The side-scroller-like *Platforms* environment requires the agent to trigger the movement of a platform just at the right time to be able to use it later on to reach the goal. Hence, the timing of the agent's actions has a lasting impact on the dynamic elements in the environment which renders credit assignment in terms of primitive actions difficult. The Platforms environment is therefore – despite its simplicity – quite challenging for existing state-of-the-art methods both "flat" and hierarchical.

**Drawbridge.**  The agent in the Drawbridge environment has to unfurl the sails of a sailing ship at the right time to pass a drawbridge immediately after it opened. Note that the agent cannot actively decelerate the ship. It is therefore essential to not accelerate too early in order not to collide with the drawbridge and lose all momentum. While this control problem is quite simple, it requires the agent to wait – an ability existing subgoal-based HRL methods do not intrinsically have.

**Tennis2D.**  In the *Tennis2D* environment a two-dimensional robot arm with a racket as an end effector has to return a ball to a specified area on the ground. As it is only through the contact between the racket and the ball that the agent can influence the outcome of the episode, it is crucial that the agent learns to precisely time the movement of the arm.

### 4.1    Empirical evaluation and comparison to baselines

We compare HiTS with two hierarchical and one "flat" baseline algorithm. As a subgoal-based HRL baseline we consider a two-level Hierarchical Actor-Critic (HAC) [19] hierarchy due to its capacity for concurrent learning and its sample efficiency. HAC uses a shortest path objective with the reward specified in Eq. 2 on both levels as well as hindsight action relabeling and HER. As a second baseline, we consider a two-level HAC hierarchy with an observation and a subgoal space that have been augmented with time. The observation is thus given by the state and the time that has passed in the current episode, $(s, t) \in \mathcal{S} \times \mathbb{N}_{\geq 0}$. An augmented subgoal $(g, \bar{t}) \in \mathcal{G} \times \mathbb{N}_{\geq 0}$ is achieved if the state $s^c$ of the agent is sufficiently close to $g$ and the current time $t$ is close to the desired achievement time $\bar{t}$. The subgoals used by this baseline hierarchy consequently contain information about when they are to be achieved. The underlying HAC algorithm is left completely unchanged, however. We furthermore consider SAC in combination with HER as a non-hierarchical baseline.

---

[2]The environments comply with OpenAI's gym interface and are available at https://github.com/martius-lab/HiTS.

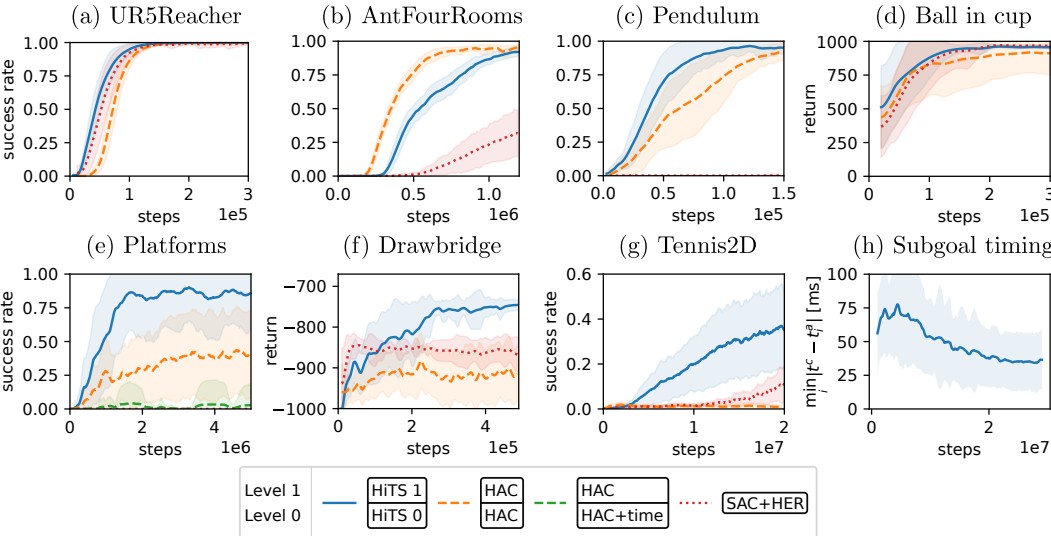

Figure 4: (a)–(g) Results on the environments. We compare HiTS against HAC, HAC with time added to the goal space and SAC. Shaded regions indicate one standard deviation over results obtained with multiple random seeds. (h) Absolute discrepancy between the time $t^c$ of contact between ball and racket and the closest subgoal achievement time $t_i^a$ in an episode of Tennis2D with HiTS. One time step in the environment corresponds to 10 ms.

In order not to bias our comparison by the choice of underlying flat RL algorithm, we use our own implementation of HAC which is based on SAC. Hence, all considered hierarchies use the same flat RL algorithm. Implementation details for HAC and HiTS can be found in Suppl. B. Details about training and hyperparameter optimization are given in Suppl. C. Note that all results reported in this section were obtained using deterministic policies that output the mean of the action distribution on each level. The performance of the stochastic policies used during training is shown in Suppl. C.

Fig. 4 (a), (b) and (c) summarize the results on three sparse-reward, long-horizon tasks introduced in [19]. Note that our implementation of HAC consistently exceeds the original results reported in [19]. HiTS furthermore outperforms HAC on two of these static environments HAC was designed for. We hypothesize that in some environments the lower level might be able to generalize over $\Delta t$ instead of having to learn how to speed up a movement by trial and error. In the *Ball in cup* environment [30] the agent has to catch a ball attached to a cup via a string. This environment can be considered dynamic as the ball is not under the direct control of the agent. Since the influence of the cup on the ball is not necessarily sparse in time and the reward depends on the state of the cup, the environment violates the assumptions of Proposition 2. Nevertheless, HiTS is able to solve the task slightly faster than the strong SAC baseline while HAC suffers from a considerable variance (see Fig. 4 (d)). Thus, in practice the second-order dynamics of an environment may allow for sufficiently tight control over a continuous interaction between agent and environment with a limited number of timed subgoals. In summary, the performance of HiTS is competitive on standard benchmark tasks, even in static environments.

Fig. 4 (e) shows the average success rates of all policy hierarchies over the course of training for the Platforms environment. The SAC baseline fails to make any progress due to the lack of structured exploration and temporal abstraction. The two-level HAC hierarchy, on the other hand, begins to learn how to solve the task but stagnates at an average success rate of around 40%. Interestingly, augmenting the observation and subgoal space with absolute time does not improve the performance of the two-level HAC hierarchy but impedes it instead. As HAC is not aware of the significance of the time component, it continues to pursue an augmented subgoal even if its time component already lies in the past. As a consequence, the agent gets "stuck" on a missed augmented subgoal until the action budget of the lower level is exhausted. As this throws timing off completely and reintroduces a non-stationarity in the induced SMDP, the solutions found by the augmented HAC hierarchy are extremely brittle. Moreover, conditioning the low-level policy on absolute time introduces a spurious explicit time dependence. In contrast to this, HiTS is adapted to the use of timed subgoals: It

conditions the lower level on the *time interval* that is left until the next timed subgoal is to be achieved. Additionally, it always queries the higher level for a new timed subgoal when this time has run out and can therefore recover from missing a timed subgoal. HiTS consequently makes fast progress on the Platforms environment and reaches an asymptotic success rate of around 86%.

A look at the learning progress of individual runs of the two-level HAC hierarchy (see Suppl. C) reveals that it does learn to solve the task in some cases. However, these solutions typically deteriorate quickly due to the learning progress on the lower level and the resulting non-stationarity of the SMDP as discussed in Section 2. In contrast to this, the performance of runs using the HiTS hierarchy is much more stable due to the use of timed subgoals as discussed in Section 3.

The challenge in the Drawbridge environment is to reach the goal (the end of the river) in the shortest possible time, which requires the right timing in order not to collide with the yet unopened bridge. Fig. 4 (f) shows the average return of all algorithms on the Drawbridge environment. In a successful episode it is equal to minus the time needed to reach the goal while an unsuccessful episode corresponds to a value of -1000 (the maximum episode length). While SAC quickly learns to immediately unfurl the sails, its local exploration in action space never finds the optimal timing. The two-level HAC hierarchy cannot reproduce the ideal behavior of waiting before unfurling the sails after the lower level has become fast at achieving subgoals and therefore stagnates at an even lower return. Increasing the subgoal budget would alleviate this problem but decrease temporal abstraction. Hence, a recursively optimal [4] HAC hierarchy may be far from optimal with respect to the environment MDP if the subgoal space is not equal to the full state space. HiTS does not suffer from this problem as the environment dynamics are independent of the agent's state in this case. Conditioning the lower level on time is therefore sufficient for HiTS to learn to time the passage through the drawbridge correctly.

In contrast to the environments discussed so far, controlling the agent in the Tennis2D environment is challenging as actions contain raw torques applied to the joints of the robot arm. Moreover, the trajectory and spin of the ball vary considerably from episode to episode. Nevertheless, HAC initially manages to return a small fraction of the balls to the goal area (see Fig. 4 (g)). Its performance decreases, however, after about 1.5 million time steps as the lower level has become too fast and a short sequence of subgoals cannot reproduce precisely timed movements anymore. As assigning credit for returning a ball to torques is challenging, SAC only learns slowly. The HiTS hierarchy can directly assign credit to a timed subgoal, i.e., to what robot state was to be achieved when. Fig. 4 (h) shows how the discrepancy between the moment of contact between ball and racket and the closest achievement time of a timed subgoal decreases during training. The higher level can therefore exercise full control over the robot arm when it matters. Moreover, the alignment of a timed subgoal with the contact between racket and ball implies that the induced SMDP is close to stationary. As a consequence, HiTS achieves a performance of around 40% on average within 20 million time steps. Also note that HiTS uses more timed subgoals around the time of contact between racket and ball as shown in Fig. S3. Thus, its temporal abstraction is adapted to the task.

The success rates of HiTS on Platforms and Tennis2D depicted in Fig. 4 (e) and (g) exhibit a large variance over random seeds. This can mostly be attributed to the environments themselves in which a small change in the behavior of the agent can have a big impact on its performance. In contrast to this, the variance of HiTS on the standard benchmarks shown in Fig. 4 (a)–(d) is much smaller. A schedule for the entropy coefficient or target entropy and the learning rate of HiTS could help with achieving proper convergence on the more challenging dynamic environments.

## 5 Related work

The key role of abstraction in solving long-horizon, sparse-reward tasks has been recognized early on in the RL community [8, 28, 25, 29, 9]. The options framework [29, 26] formalizes the idea of augmenting the action space with closed-loop activities that take over control for an extended period of time. The resulting temporal abstraction facilitates credit assignment for a policy selecting among options. While options were originally handcrafted or learned separately [4, 26], the option-critic architecture [3] jointly learns the parameters of the options and a policy over options. However, to keep up stable temporal abstraction in this approach, regularization is necessary [3, 14]. A related approach to learning a hierarchy is based on the idea that a high-level controller should have a repertoire of skills to choose from [7] or to modulate [12, 15, 11, 10]. The set of skills is often

pretrained using an auxiliary reward where diversity is encouraged by an information theoretic term [11, 10].

Alternatively, the lower level of the hierarchy may be trained to reach an assigned subgoal [27, 8, 26, 22]. In this setting, the lower level can be rewarded for moving in a specified direction in goal space [31], for being close to the goal state [23] or for achieving a goal state up to a tolerance [19]. In the latter case it is natural to query the higher level for a new subgoal only when the current one has been achieved or a fixed action budget is exhausted [19]. In contrast to this, using a dense reward is usually tied to a fixed temporal abstraction scheme [23, 31]. The goal space can either be learned [31, 24] or predefined using domain knowledge [23, 19].

It has been observed in the context of options [29, 26] as well as subgoal-based methods [23, 19] that the higher level in a hierarchy effectively interacts with a non-stationary SMDP. In order to enable off-policy learning from stored high-level transitions, Nachum et al. [23] approximately determine high-level actions that are likely to reproduce the stored behavior of the lower level under the current low-level policy. This procedure has to be repeated after each update to the low-level policy and introduces noise due to the use of sampling in the approximation. Levy et al. [19] address the non-stationarity issue by hindsight action relabeling as discussed in Section 2. None of these methods take into account, however, that, when using adaptive temporal abstraction as in [19], the transition times of the SMDP change over the course of training and consequently introduce an additional source of non-stationarity which prevents sample-efficient learning in dynamic environments. Although in Blaes et al. [5] the environment contains dynamic objects, they come to rest quickly if not manipulated by the agent, such that the non-stationarity issue does not need to be addressed.

As hindsight action relabeling hides the limitations of the current low-level policy, the higher level may learn to choose infeasible subgoals. Levy et al. [19] therefore penalize the higher level when the lower level fails to reach a subgoal. We adopt this method as it is general and simple. Zhang et al. [33] restrict the subgoal to a k-step adjacent region via an adjacency loss. As the loss is based on an adjacency network distilled from an adjacency matrix, incorporating this approach into HiTS may improve sample efficiency, albeit only on low-dimensional subgoal spaces as it requires discretization.

In the context of robotics, via-points are used to specify a sequence of configurations a robot should move through, often also specifying when these configurations are to be reached [21]. They therefore serve a similar purpose as timed subgoals. The difference lies in how they are used. A classic approach in robotics would be to first obtain a controller for the robot, to then specify via-points, interpolate between them, and use the controller to follow the resulting trajectory. Hence, no concurrent learning is involved and the non-stationarity problem does not occur.

## 6 Conclusion

We present an HRL algorithm designed for sample-efficient learning in dynamic environments. In a theoretical analysis, we show how its use of timed subgoals in conjunction with hindsight action relabeling attenuates the non-stationarity problem of HRL even when the lower level is not conditioned on the full state. Moreover, our experiments demonstrate that our method is competitive on a range of standard benchmark tasks and outperforms existing state-of-the-art baselines in terms of sample complexity and stability on three new challenging dynamic tasks.

The effectiveness of HRL generally depends on the structure of the environment and our method is no exception in this regard. In particular, we consider environments with a directly controllable part which only sparsely influences the rest. This assumption ensures that temporal abstraction is compatible with stable concurrent learning. Furthermore, our practical algorithm still contains minor sources of non-stationarity like testing transitions. Nevertheless, our experiments demonstrate that our method performs well even when deviating from the idealized setting.

An interesting direction for future work is the detection of interactions between agent and environment and the active alignment of timed subgoals to them. Furthermore, two or more levels which pursue timed subgoals may enable efficient learning on more challenging dynamic long-horizon tasks.

## Acknowledgments and Disclosure of Funding

We would like to thank Bernhard Schölkopf, Jan Peters, Alexander Neitz, Giambattista Parascandolo, Diego Agudelo España, Hsiao-Ru Pan, Simon Guist, Sebastian Gomez-Gonzalez, Sebastian Blaes and Pavel Kolev for their valuable feedback.

Georg Martius is a member of the Machine Learning Cluster of Excellence, EXC number 2064/1 – Project number 390727645. We acknowledge the support from the German Federal Ministry of Education and Research (BMBF) through the Tübingen AI Center (FKZ: 01IS18039B).

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
