# Supplementary Material:
# Hierarchical Reinforcement Learning
# With Timed Subgoals

## A   Proofs

In this section we provide proofs for Prop. 1 and 2. For the sake of clarity, we denote random variables by capital letters and realizations of random variables by lowercase letters. The probability density of a random variable X conditioned on another random variable Y is denoted by $p_{X|Y}(x \mid y)$.

Prop. 1 is concerned with the distribution of high-level transitions generated by hindsight action relabeling [19],

$$(s_t, a_t^1, r_t, s_{t+\tau}) \implies (s_t, \hat{a}_t^1 := s_{t+\tau}, \hat{r}_t^1 := r^1(s_t, \hat{a}_t^1), s_{t+\tau}) \quad \text{(hindsight action relabeling)}, \quad (8)$$

where $\tau$ denotes the transition time of the high-level SMDP, i.e., how long the lower level stays in control.

**Proposition 1** *Applying hindsight action relabeling (Eq. 8) at level 1 generates transitions that follow a stationary state transition and reward distribution, i.e. $p_t\left(s, \hat{a}^1, \hat{r}^1, s'\right) = p_t\left(s, \hat{a}^1\right) p\left(s', \hat{r}^1 \mid s, \hat{a}^1\right)$ where $p\left(s', \hat{r}^1 \mid s, \hat{a}^1\right)$ is time-independent provided that the subgoal space is equal to the full state space.*

*Proof.* For clarity we omit the superscript 1 in the proof as actions and rewards always correspond to the higher level. Consider random variables $S_t$, $S_{t+\tau}$ representing the current and next state in a transition. Both take values in the state space $\mathcal{S}$ and may have an arbitrary time-dependent joint density function $p_{S_t, S_{t+\tau}}(s_t, s_{t+\tau})$, i.e., in general $p_{S_s, S_{s+\tau}} \neq p_{S_t, S_{t+\tau}}$ for $s \neq t$. Applying hindsight action relabeling Eq. 8 then corresponds to defining $\hat{A}_t = S_{t+\tau}$ and $\hat{R}_t = r^1(S_t, \hat{A}_t)$. The joint probability density of the next state $S_{t+\tau}$ and the relabeled reward $\hat{R}_t$ conditioned on the state $S_t$ and the relabeled action $\hat{A}_t$ is then given by

$$p_{S_{t+\tau}, \hat{R}_t | S_t, \hat{A}_t}\left(s_{t+\tau}, \hat{r}_t \mid s_t, \hat{a}_t\right) = p_{\hat{R}_t, \hat{A}_t | S_t, S_{t+\tau}}\left(\hat{r}_t, \hat{a}_t \mid s_t, s_{t+\tau}\right) \frac{p_{S_t, S_{t+\tau}}\left(s_t, s_{t+\tau}\right)}{p_{S_t, \hat{A}_t}\left(s_t, \hat{a}_t\right)} \quad (9)$$

$$= \delta\left(\hat{r}_t - r^1\left(s_t, \hat{a}_t\right)\right) \delta\left(\hat{a}_t - s_{t+\tau}\right) \frac{p_{S_t, S_{t+\tau}}\left(s_t, s_{t+\tau}\right)}{p_{S_t, \hat{A}_t}\left(s_t, \hat{a}_t\right)} \quad (10)$$

$$= \delta\left(\hat{r}_t - r^1\left(s_t, \hat{a}_t\right)\right) \delta\left(\hat{a}_t - s_{t+\tau}\right) \frac{p_{S_t, S_{t+\tau}}\left(s_t, s_{t+\tau}\right)}{p_{S_t, S_{t+\tau}}\left(s_t, \hat{a}_t\right)} \quad (11)$$

$$= \delta\left(\hat{r}_t - r^1\left(s_t, \hat{a}_t\right)\right) \delta\left(\hat{a}_t - s_{t+\tau}\right) \frac{p_{S_t, S_{t+\tau}}\left(s_t, s_{t+\tau}\right)}{p_{S_t, S_{t+\tau}}\left(s_t, s_{t+\tau}\right)} \quad (12)$$

$$= \delta\left(\hat{r}_t - r^1\left(s_t, \hat{a}_t\right)\right) \delta\left(\hat{a}_t - s_{t+\tau}\right) , \quad (13)$$

where $\delta$ denotes the Dirac delta function. In (11) we have used $\hat{A} = S_{t+\tau}$. Hence, the conditional distribution of the next state and the reward given the current state and the action is stationary. $\square$

Prop. 2 extends this result to the transition time of the SMDP given that timed subgoals and hindsight action relabeling are used,

$$(s_t, a_t^1 = (g_t^0, \Delta t_t^0), r_t, s_{t+\Delta t_t^0}) \implies (s_t, \hat{a}_t^1 := (\hat{g}_t^0, \Delta t_t^0), \hat{r}^1 := r^1(s_t, \hat{a}_t^1), s_{t+\Delta t_t^0}) . \quad (14)$$

**Proposition 2** *If the not directly controllable part of the environment evolves completely independently of the controllable part, i.e., if $p\left(s'^e \mid s^e\right) = p(s'^e \mid s^e, s^c, a^0)$, and if hindsight action relabeling is used, the transitions in the replay buffer of a higher level assigning timed subgoals to a lower level are consistent with a completely stationary SMDP. Thus, they follow a distribution $p_t\left(s, \hat{a}^1, \tau, s', \hat{r}^1\right) = p_t\left(s, \hat{a}^1\right) p\left(s', \tau, \hat{r}^1 \mid s, \hat{a}^1\right)$ where $p\left(s', \tau, \hat{r}^1 \mid s, \hat{a}^1\right)$ is time-independent.*

*Proof.* We again omit the superscript 1 in the proof as actions and rewards always correspond to the higher level. In addition to $S_t$, $S_{t+\tau}$ as defined in the proof of Prop. 1, consider random variables

$\Delta T_t$ and $G_t$ corresponding to the desired time until achievement of a timed subgoal and the subgoal, respectively. Recall that the state is assumed to consist of a controllable and a non-controllable part, $S_t = (S_t^c, S_t^e)$. Note that applying hindsight action relabeling (Eq. 14) then corresponds to defining $\hat{G}_t = S^c{}_{t+\tau}$. For the sake of brevity we drop the specification of the random variables in the subscript of the density, i.e., $p(x, y) := p_{X,Y}(x, y)$. Consider the probability density of the next state, the transition time and the reward given the current state, the subgoal and the desired time until achievement,

$$p\left(s_{t+\tau}, \tau, \hat{r}_t \mid s_t, \hat{g}_t, \Delta t_t\right) = p\left(\tau, \hat{r}_t, \hat{g}_t \mid s_t, s_{t+\tau}, \Delta t_t\right) \frac{p^t\left(s_t, s_{t+\tau}, \Delta t_t\right)}{p^t\left(s_t, \hat{g}_t, \Delta t_t\right)} \tag{15}$$

$$= \delta\left(\tau - \Delta t_t\right) \delta\left(\hat{r}_t - r^1\left(s_t, \hat{g}_t, \Delta t_t\right)\right) \delta\left(\hat{g}_t - s_{t+\tau}^c\right) \tag{16}$$

$$\cdot \frac{p^t\left(s_t, s_{t+\tau}, \Delta t_t\right)}{p^t\left(s_t, \hat{g}_t, \Delta t_t\right)} \tag{17}$$

$$= \delta\left(\tau - \Delta t_t\right) \delta\left(\hat{r}_t - r^1\left(s_t, s_{t+\tau}^c, \Delta t_t\right)\right) \delta\left(\hat{g}_t - s_{t+\tau}^c\right) \tag{18}$$

$$\cdot \frac{p^t\left(s_t, s_{t+\tau}, \Delta t_t\right)}{p^t\left(s_t, s_{t+\tau}^c, \Delta t_t\right)} , \tag{19}$$

where we used the superscript $t$ to highlight that a probability density is time-dependent. Since the time evolution of $s^e$ was assumed to be independent of $s^c$ and $a$, the fraction is time-independent,

$$\frac{p^t\left(s_t, s_{t+\tau}, \Delta t_t\right)}{p^t\left(s_t, s_{t+\tau}^c, \Delta t_t\right)} = \frac{p^t\left(s_{t+\tau} \mid s_t, \Delta t_t\right) p^t\left(s_t, \Delta t_t\right)}{p^t\left(s_{t+\tau}^c \mid s_t, \Delta t_t\right) p^t\left(s_t, \Delta t_t\right)} \tag{20}$$

$$= \frac{p\left(s_{t+\tau}^e \mid s_t, \Delta t_t\right) p^t\left(s_{t+\tau}^c \mid s_t, \Delta t_t\right) p^t\left(s_t, \Delta t_t\right)}{p^t\left(s_{t+\tau}^c \mid s_t, \Delta t_t\right) p^t\left(s_t, \Delta t_t\right)} \tag{21}$$

$$= p\left(s_{t+\tau}^e \mid s_t, \Delta t_t\right) . \tag{22}$$

Hence, the conditional density

$$p\left(s_{t+\tau}, \tau, \hat{r}_t \mid s_t, \hat{g}_t, \Delta t_t\right) = \delta\left(\tau - \Delta t_t\right) \delta\left(\hat{r}_t - r^1\left(s_t, \hat{g}_t, \Delta t_t\right)\right) \delta\left(\hat{g}_t - s_{t+\tau}^c\right) p\left(s_{t+\tau}^e \mid s_t, \Delta t_t\right) \tag{23}$$

corresponds to a completely stationary SMDP. $\square$

## B  Algorithm implementation details

In this section we discuss our implementations of HAC and HiTS. Both support Deep Deterministic Policy Gradient (DDPG) [20] and Soft Actor-Critic (SAC) [13] as underlying off-policy reinforcement learning algorithms. We used the implementation of SAC and DDPG provided by the reinforcement learning library Tianshou [32]. For our experiments in Section 4 we only used SAC, however, as it performed better in tests on the Platforms environment.

### B.1  Hierarchical Actor-Critic (HAC)

Our implementation of HAC deviates slightly from the one presented in Levy et al. [19] in order to improve performance and flexibility:

- We use SAC in instead of DDPG in our experiments.
- We do not squash the learned Q-functions to the interval $[-H, 0]$ where $H > 0$ denotes the maximum number of actions on a level until control is returned to the next higher level or the episode ends. Instead, we apply the mapping

$$f(x) = \log\left(\frac{1}{1 + \exp(-x)}\right)$$

  to the output of a multilayer perceptron to map it to $(-\infty, 0)$. This ensures that the learned Q-function can reach values below $-H$ which occur in the true Q-function due to bootstrapping. Furthermore, this formulation allows for adding negative auxiliary rewards like the entropy term of SAC or regularization terms.

- In contrast to [19] we bootstrap when learning from failed testing transitions, i.e., we do not set the discount rate to zero. We do this to prevent failing earlier from being more attractive to the agent than failing later. In other words, if no bootstrapping was used, the higher level could avoid credit assignment for negative rewards occurring later in the episode by immediately choosing an infeasible subgoal. If the current low-level policy is likely to not achieve an assigned subgoal at some point during the episode, it would make sense for the higher level to immediately choose an infeasible subgoal.

- Our implementation is compatible with environments complying with OpenAI's goal-based gym interface [6].

- We use a more flexible version of HER than Levy et al. [19]. In particular, we not only implement the "episode" hindsight goal sampling strategy but also the "future" variant and we do not always use the last state of the episode as a hindsight goal. Furthermore, we allow for custom filtering of the achieved goals in an episode before sampling from them.

### B.2 Hierarchical reinforcement learning with timed subgoals (HiTS)

The implementation details mentioned in Section B.1 also apply to the higher level in the HiTS hierarchy which shares code with our HAC implementation.

Moreover, we restricted the desired time until achievement $\Delta t_t^0$ the higher level outputs to an interval $I = (0, \Delta t^{\max})$ with $\Delta t^{\max} > 0$ in order to prevent the higher level from offloading the complete task to the lower level. Furthermore, we sample a fixed fraction (5% in our experiments) of all timed subgoals from a uniform distribution over $\mathcal{G}^0 \times I$ in order to improve exploration and make sure that the replay buffer contains penalties for all infeasible regions of the goal space similar to Levy et al. [19].

In practice, we let the high-level policy output a real-valued desired time until achievement $\Delta t_t^0 \in I$. We then decrement in each time step, $\Delta t_{t+1}^0 = \Delta t_t^0 - 1$, and check for $\Delta t_{t+1}^0 \leq 0$ as a condition for querying a new timed subgoal from the higher level. This is equivalent to first rounding to the next greater integer and then checking for equality to zero, i.e., $\lceil \Delta t_{t+1}^0 \rceil = 0$. The notation $\Delta t_t^0 \in \mathbb{N}_{>0}$ we chose in Section 3.1 is therefore compatible with our implementation based on real-valued $\Delta t_t^0$ but easier to parse.

Note that the choice of SAC as an underlying off-policy algorithm facilitates the alignment of the desired achievement times $t + \Delta t_t^0$ with those points in time when the controllable part interacts with the rest of the environment. See Section C.7 for a discussion of this mechanism.

## C Experiments

### C.1 Environments and hierarchies

If not stated otherwise, the observation space of the lower level and the subgoal space are the state space of the agent. The architecture of the hidden layers of the policies and Q-functions are the same over algorithms.

**Platforms.** As an exception, in the Platforms environment all levels see the full state. Thus, the lower level has a fair chance of learning to cope with the dynamic elements of the environment even if no timing information is communicated by the higher level. In particular, the lower level of the HAC hierarchy can learn not to fall off the lower platform by waiting for the moving platform before trying to roll onto it.

**Tennis2D.** When applying HER on the higher level of all hierarchies on the Tennis2D environment, only achieved states which correspond to the second contact between ball and ground are considered. This choice is motivated by the observation that generalization from arbitrary states of the environment to goals corresponding to the ball bouncing of the ground might be difficult. As all hierarchies and the non-hierarchical baseline use this version of HER, it does not bias the results in Section 4.

**Ball in cup.** As this environment is not goal-based, HAC was modified to use the reward specified in Eq. 4. The higher level of the hierarchy is therefore quite similar to its counterpart in HiTS.

Nevertheless, HAC suffers from a significantly higher variance on this environment which can be attributed to the use of subgoals as opposed to timed subgoals.

## C.2  Training and hyperparameter optimization

Table S1: Summary of hyperparameter optimization.

|  | **UR5Reacher** | **AntFourRooms** | **Pendulum** | **Ball in cup** |
|---|---|---|---|---|
| objective | success rate | success rate | success rate | return |
| average over steps | $\{0, ..., 5e5\}$ | $\{0, ..., 1.2e6\}$ | $\{0, ..., 5e5\}$ | $\{0, ..., 5e5\}$ |
| trials | 40 | 100 | 60 | 60 |
| parallel trials | 20 | 50 | 20 | 20 |
| seeds per trial | 5 | 10 | 10 | 10 |
|  | **Platforms** | **Drawbridge** | **Tennis2D** |  |
| objective | success rate | return | success rate |  |
| average over steps | $\{1e6, ..., 3e6\}$ | $\{0, ..., 5e5\}$ | $\{5e6, ..., 1e7\}$ |  |
| trials | 60 | 40 | 20 |  |
| parallel trials | 20 | 20 | 20 |  |
| seeds per trial | 10 | 7 | 10 |  |

The hyperparameter optimization was conducted using the Optuna framework [1] with the Tree-structured Parzen Estimator sampler. The objective was to maximize the average success or return over a fixed range of time steps. Since performance on the more challenging environments is quite stochastic, multiple seeds were ran per trial. To keep the overall runtime low, 20 trials were ran in parallel. Table S1 summarizes the hyperparameter optimization on all considered environments.

The results of the best trial of an algorithm on an environment were then reproduced with different random seeds to remove the maximization bias. All results reported in Fig. 4 are obtained from 30 different seeds, except for HAC and SAC on Tennis2D where only 25 and 5 seeds were considered, respectively. The remaining 5 and 25 seeds led to runs that diverged at some point after $1e7$ time steps.

Unless otherwise specified, the hyperparameters which were optimized are:

- learning rate (same for all levels)
- target smoothing coefficient (same for all levels)
- entropy coefficient mode (fixed or tuned according to target entropy) (per level)
- entropy coefficient or target entropy (per level)
- $c$ from Eq. 4 (only HiTS level 1)

The remaining fixed hyperparmeters (like the subgoal budget) were the same across algorithms. In case of the AntFourRooms environment separate learning rates for both levels and the timed subgoal budget were optimized for HAC and HiTS which improved the performance of both algorithms. We refer to the configuration files for a list of the optimized hyperparameters.

## C.3  Hardware and runtimes

The experiments presented in Section 4 and Appendix C were run on a cluster with several different CPU models. For concreteness, we give approximate runtimes per random seed on one core of an Intel Xeon IceLake-SP 8360Y in Table S2. We only list the runtimes of our HiTS implementation. The runtimes of all baselines were similar, however.

## C.4  Results for stochastic policies

Fig. S1 shows the performance of the stochastic policies used during training as a function of steps taken (in contrast to Fig. 4 which shows learning curves for deterministic policies outputting the mean of the action distribution). Due to the exploration in action space, the success rates and returns of the stochastic policies are usually lower than those of their deterministic counterparts.

Table S2: Runtimes per random seed for HiTS on one core of an Intel Xeon IceLake-SP 8360Y.

| | Platforms | Drawbridge | Tennis2D | UR5Reacher | AntFourRooms |
|---|---|---|---|---|---|
| time steps | 5e6 | 5e5 | 2e7 | 5e5 | 1.2e6 |
| runtime [h] | 12 | 0.5 | 28 | 0.7 | 1.6 |

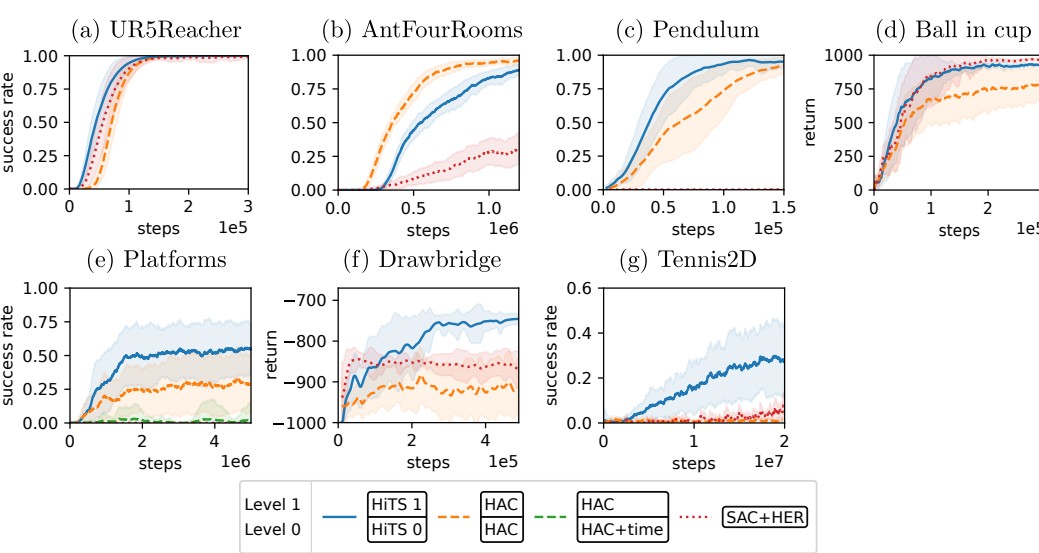

Figure S1: Results on the environments with the stochastic policies used during training. We compare HiTS against HAC, HAC with time added to the goal space and SAC. Shaded regions indicate one standard deviation over results obtained with multiple random seeds.

## C.5 Analysis of individual runs on the Platforms environment

Fig. S2 shows the success rates of HAC and HiTS for five different seeds on the Platforms environment. While HAC manages to find a solution for some seeds, it usually deteriorates quickly due to the learning progress on the lower level and the resulting non-stationarity of the SMDP the higher level interacts with (as discussed in Section 4). As a result, the performance of the HAC hierarchies is quite unstable. HiTS, on the other hand, quickly finds a solution for all random seeds. For some seeds the performance drops again for a limited amount of time steps. This can be attributed to the remaining non-stationarity of the effective SMDP as discussed in Section 3.1 as well as to the task itself for which the optimal policy is very close to behavior that leads to complete failure.

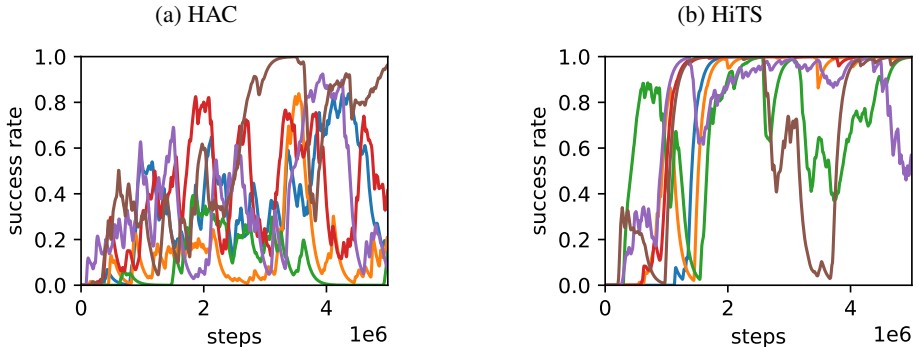

Figure S2: Five individual runs with different random seeds on the Platforms environment generated with a two-level HAC hierarchy (a) and a HiTS hierarchy (b).

## C.6 Distribution of subgoal achievement times on Tennis2D

While the ball is still far away from the racket in the Tennis2D environment, it is not necessary for the agent to be reactive. It can therefore afford temporal abstraction in the beginning of the episode. As soon as the ball gets close to the racket, however, the agent might benefit from being able to adjust its actions based on the current position, velocity and spin of the ball. Thus, the sweet spot in the trade-off between temporal abstraction and reactiveness changes during an episode of Tennis2D.

This is reflected in the way HiTS distributes timed subgoals over the episode. Fig. S3 shows a histogram of the difference between subgoal achievement times and the time of the contact between ball and racket for HiTS and HAC. While HiTS uses more timed subgoals around the time of the contact, HAC distributes its subgoals more uniformly over the episode. This indicates that the higher level of the HiTS hierarchy has learned to choose small $\Delta t$ when it is crucial for performance and larger ones when it can afford it. The explicit representation of the time interval between consecutive timed subgoals can be expected to facilitate HiTS' adaptation of temporal abstraction to the environment.

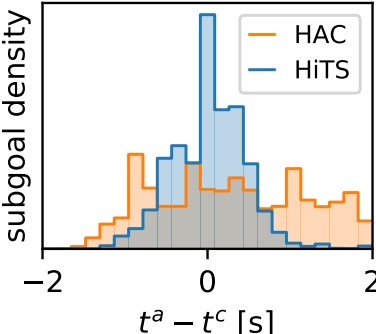

Figure S3: Histogram of the difference between subgoal achievement times $t^a$ and the time $t^c$ of the contact between racket and ball in the Tennis2D environment. HiTS uses more timed subgoals around the contact whereas HAC distributes its subgoals more uniformly over the episode. Averaged over 30 seeds.

## C.7 Stochasticity of the low-level policy while pursuing a timed subgoal

In Section 3 we discussed the assumption that the higher level learns to line up timed subgoals with those points in time at which the agent influences the rest of the environment (see Fig. 2 (a)). Fig. 4 (h) demonstrates that this happens in practice in the Tennis2D environment (at least up to a time scale on which the inertia of the robot arm renders the influence of low-level actions insignificant). This alignment is incentivized by the use of SAC: The lower level, which is trained using a maximum entropy objective, is particularly stochastic inbetween timed subgoals as it can afford to explore there. When $\Delta t$ is close to 0, on the other hand, the noise on the agent's state is small since this is necessary for achieving the timed subgoal (see Fig. S4). Hence, the higher level is encouraged to align the subgoal achievement times with those points in time when agent and environment interact so as to solve the task unimpeded by low-level noise. It is an interesting direction for future work that could improve sample efficiency to more actively guide the alignment of timed subgoals with interactions between agent and environment.

## C.8 Influence of the (timed) subgoal budget on performance

As the (timed) subgoal budget was not optimized except in the AntFourRooms environment, we investigated its influence on the performance of HiTS and HAC on the Platforms task. Fig. S5 shows the success rate of both algorithms as a function of the (timed) subgoal budget. The maximum number of actions on the lower level (HAC) or the maximum $\Delta t$ (HiTS) was scaled inversely proportional to the budget, all other hyperparameters were left fixed. The performance of HAC improves slightly with the subgoal budget but stagnates at around 50%. While HiTS profits from a slightly increased

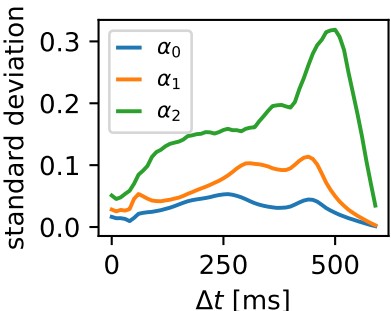

Figure S4: Standard deviation of the joint angles (in radians) of the robot arm from the Tennis2D environment as a function of $\Delta t$. The data was generated by initializing the robot arm in a fixed position and letting the lower level pursue the same timed subgoal 100 times.

timed subgoal budget performance suffers when increasing the budget further. This can be attributed to the simultaneous inverse scaling of the maximum $\Delta t$ which reduces temporal abstraction. In summary, the (timed) subgoal budget does have an influence of the performance of both algorithms but even when optimizing over it, HAC still cannot solve the Platforms environment.

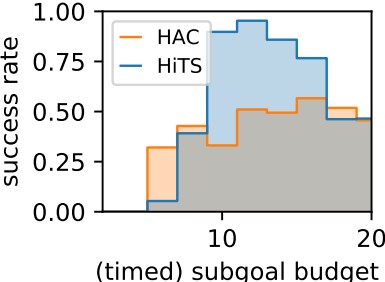

Figure S5: (Timed) subgoal budget sweep on the Platforms environment. The (timed) subgoal budget used for the experiments in the Platforms environment presented in the main text was 10. Success rates averaged over 30 seeds.