# OpenReview forum: "Hierarchical Reinforcement Learning with Timed Subgoals"
_NeurIPS.cc/2021/Conference — NeurIPS 2021 Poster_

### Official Review · Reviewer_Auot · 2021-07-16

**Rating:** 6
**Confidence:** 3

**Summary:**

The authors introduce HiTS, an HRL algorithm in which the high-level policy communicates not only which goal to achieve to the low-level policy but also when to achieve it. This feature enables the agent to adapt to dynamic environments that require planning over long horizons. The authors also discuss to what extent communicating timed sub-goals results in more stable learning for the high-level policy. The authors show that HiTS successfully learn on a set of 3 benchmarks, that have been designed to be dynamic and require planning, where classical sub-goals HRL method fail to learn stable solutions.

**Limitations And Societal Impact:**

This method does not add limitations or potential negative societal impact to existing reinforcement learning methods.

**Main Review:**

I warmly thank the authors for their work.

Soundness of the claims, significance and novelty of the contribution, and relevance to the NeurIPS community:

- The authors motivate clearly the need for timed sub-goals, when the environment is dynamic, through examples.

- The work is well-positioned with respect to the literature.

- The 3 benchmarks are sound with respect to the problem at hand in this paper. HiTS clearly outperforms HAC in these environments, motivating the need for timed sub-goals in this setting.

Limitations of this work:

- At the beginning of the introduction, the authors argue that to realize the full potential of HRL, it is necessary to enable concurrent learning on all levels of hierarchy and present HiTS as a way to do so. They argue that timed sub-goals reduces the non-stationarity of the induced SMDP and thus stabilize learning for any HRL problem. However, in the rest of the paper, the method seemed to be designed specifically for dynamic environments. I think it would be important either to focus on dynamic environments only or to spend more time validating that timed sub-goals is important for any HRL problem, i.e for environments that require planning over long time horizons but are static.

- In the appendix, the authors show that HiTS is clearly outperformed by HAC on the AntFourRooms environment which interrogates whether timed sub-goals are also important when the environment is not dynamic. It would be interesting to further discuss this point in the paper.

- The three environments might be more carefully detailed in the experimental sections. Notably, the observation space, action space, goal space, and reward functions should be more precisely described.

- The second HAC baseline for which the high-level policy communicates timed sub-goals to the low-level policy remains unclear to me. I do not get precisely what is the difference with HiTS. I think it would be worth allocating space in the paper to precise the difference between the two methods.

- It is also unclear to me how the time interval is communicated between the two levels. Is it under the form of a scalar or a one-hot representation? In general, the different components of the agent and the neural architecture used might be more detailed with maybe explicating figures.

Clarity:

- The paper is well written and overall easy to follow.


I think that in this form this paper is marginally below the acceptance threshold but I am willing to increase my score if the authors address the limitations I mentioned.

**Time Spent Reviewing:**

4

---

> ### Author Response · Authors · 2021-08-10
> **Response to Auot**
>
> Thank you very much for your helpful review.
>
> > At the beginning of the introduction, the authors argue that to realize the full potential of HRL, it is necessary to enable concurrent learning on all levels of hierarchy and present HiTS as a way to do so. They argue that timed sub-goals reduces the non-stationarity of the induced SMDP and thus stabilize learning for any HRL problem. However, in the rest of the paper, the method seemed to be designed specifically for dynamic environments. I think it would be important either to focus on dynamic environments only or to spend more time validating that timed sub-goals is important for any HRL problem, i.e for environments that require planning over long time horizons but are static.
>
> > In the appendix, the authors show that HiTS is clearly outperformed by HAC on the AntFourRooms environment which interrogates whether timed sub-goals are also important when the environment is not dynamic. It would be interesting to further discuss this point in the paper.
>
> The design of HiTS is indeed targeted at dynamic environments as we mentioned in the abstract and the introduction. We believe dynamic environments are highly relevant for applications of HRL but are underrepresented in the literature. While a conventional subgoal-based HRL method induces an SMDP with transition times that decrease over the course of training, we do not think this impedes performance in static environments as briefly discussed in line 43 f. We will put more emphasis on our focus on dynamic environments in our revision, in particular when formulating our contribution.
> The experiments depicted in Fig. S1 in the Suppl. Material demonstrate that HiTS can learn tasks in static environments but timed subgoals do not lead to improved performance in this setting, as expected. Due to the restricted page count, we lean towards concentrating on dynamic environments in the main text and keeping these experiments in the Supplementary Material.
>
> > The three environments might be more carefully detailed in the experimental sections. Notably, the observation space, action space, goal space, and reward functions should be more precisely described.
>
> The reward function is identical in all environments and is discussed in line 251 ff. Additional details about the environments are given in Section C.1 in the Supplementary material. We will add a more detailed description of observation, action, and goal spaces.
>
> > The second HAC baseline for which the high-level policy communicates timed sub-goals to the low-level policy remains unclear to me. I do not get precisely what is the difference with HiTS. I think it would be worth allocating space in the paper to precise the difference between the two methods.
>
> The HAC baseline with an observation and a subgoal space that have been augmented with time illustrates that it is not enough to simply include time into a subgoal to achieve sample-efficient learning. While it was briefly mentioned in line 177 ff. and discussed in 274 ff. and 292 ff. we agree that being more explicit would improve readability and will therefore dedicate more space to it in the experiments section. We will add roughly the following material:
>
> The observation in this baseline is given by the state and the time that has passed in the current episode, $(s, t)\in \mathcal{S}\times \mathbb{N}_{\geq0}$.
>
> An augmented subgoal $(g, \bar{t})\in \mathcal{G}\times \mathbb{N}_{\geq0}$ is achieved if the state $s^c$ of the agent is sufficiently close to $g$ and the current time $t$ is close to the desired achievement time $\bar{t}$. The subgoals used by this baseline hierarchy consequently contain information about when they are to be achieved but the HAC algorithm is left unchanged.
>
> In contrast to HiTS, this baseline (i) communicates in terms of absolute time (ii) does not automatically query a new subgoal at $t=\bar{t}$ if the current one was not achieved and (iii) uses the HAC reward (eqn. above line 97) instead of eqn. (4) on the lower level. This causes several problems, most notably:
> 1. conditioning the lower level on absolute time introduces a spurious time dependence into the policy
> 2. the agent gets “stuck” on a missed subgoal until the action budget of the lower level is exhausted.
>
> The second point, in particular, is a limiting factor as this happens frequently in an episode and completely “paralyzes” the agent.
>
> > It is also unclear to me how the time interval is communicated between the two levels. Is it under the form of a scalar or a one-hot representation? In general, the different components of the agent and the neural architecture used might be more detailed with maybe explicating figures.
>
> The time interval $\Delta t$ is communicated as a scalar to the low-level policy. Details about the representation of $\Delta t$ in the implementation of HiTS can be found in Section B.2 in the Suppl. Material. Due to the limited page count, we were not able to discuss implementation details in the main text. In order to convey a more complete picture of the algorithm, we have added a figure (see Fig. +1 on the [Project page](https://sites.google.com/view/hrl-with-timed-subgoals/home)) with a detailed illustration of the execution of HiTS over time. We will include this figure in the main text.

---

> > ### Comment · Reviewer_Auot · 2021-08-27
> > **Thank you!**
> >
> > Thanks very much for the detailed response. My questions have been clarified.

---

### Official Review · Reviewer_3XHt · 2021-07-16

**Rating:** 6
**Confidence:** 3

**Summary:**

This paper presents a new framework for subgoal-based HRL where the high-level chooses not only what subgoal to be reached but also when to reach the subgoal, i.e., a timed subgoal, for the lower-level. By emitting such timed subgoals, the non-stationarity of transition times during learning at the high-level can be removed, which is supported by theoretical analysis, and thus, making it possible for stable learning in dynamic environments. Combining timed subgoals with hindsight action relabeling, a practical algorithm HiTS is proposed and it outperforms baseline methods on a set of dynamic environments.

**Limitations And Societal Impact:**

The authors discussed the limitation but did not address negative social impact. The proposed algorithm helps improve learning in complex and long-horizon dynamic environments. As an illustrative example, what if the algorithm was used by a malicious user to build a robot? Targeting rewards that lead to some harmful behaviors for a society (in a dynamic environment)? I believe such kind of scenario could be discussed. In addition, what can and should be done to prevent it from happening?

**Main Review:**

In HRL, the non-stationarity at the high-level learning is a well-known problem that hinders concurrent learning and learning efficiency. The problem gets even worse when learning in dynamic environments as fully analyzed in the paper. The proposed solution, which generates timed subgoals to the lower-level, is a new and interesting idea, highlighting the importance of communication and coordination between the two levels for effective learning in a complex environment. Overall, the paper is well-written and technically sound. However, I feel that some parts of the paper, e.g., Section 3.1, can be better organized and the experimental results could be further refined to improve the clarity and the overall quality of the paper. Please see the list of items below for detailed comments, and I am willing to adjust my evaluation should these problems be addressed or fully discussed.

(1)	Section 3.1 could be better organized to improve clarity. I spent most of the time understanding all the algorithmic details in section 3.1. It could be better to break down this section into subsections where each subsection highlights and introduces one aspect of the algorithm. More specifically,

1a.	(line 215-216) says “the lower level receives a non-zero reward only if it achieves a timed subgoal”. This is inconsistent with the equation above line 97, which assigns zero rewards to the lower-level when near the subgoal.

1b.	(line 233-235) reads “Furthermore, while pursuing … the higher level is penalized.” I feel like this part needs more elaboration.

1c.	(line 237-239) reads “While these two exceptions … respect the limitations of the lower level”. It would be better to clearly mention that this part is the limitation of the proposed algorithm.

1d.	Is there any transition function used for $g^0$ from $t$ to $t+1$? It could be added to improve clarity.

(2)	Please see the two comments below regarding experiments:

2a.	Although discussed in the paper, the high variance of HiTS in the experiments does call for further investigation and analysis. How many random seeds are used for generating the results (I don’t think it is mentioned in the paper)? Does that mean the non-stationarity in high-level learning still exists? Is the performance dependent on choosing the right seed? Can this be alleviated by conservative exploration as hypothesized in the paper?

2b.	Since the three dynamic environments are likely to be new to the readers and they all focus on learning the “right timing”, it worth evaluating and comparing HiTS on commonly-used benchmark environments as well. I would suggest bringing the additional static experiments on UR5Reacher and AntFourRooms back to the main paper, which are more familiar by the readers. Can “increasing the subgoal budget” improve the learning efficiency for HiTS on AntFourRooms? It would be interesting to see and compare results with different budgets, and see how well HiTS can perform on static benchmarks.

(3)	To my understanding, timed subgoals highlight the need for communication between high- and lower-levels for concurrent and effective learning. Can other forms of communication (partly) address the non-stationarity issue in practice? Mentioned in (line 338 – 341), generating feasible subgoals seems to be important for learning efficiency. Recently, [1] proposed an algorithm, HRAC, that generates subgoals that are k-step adjacent to the current state, improving the reachability of subgoals at the lower-level. Since the adjacency is learned by sampled trajectory data and updated periodically, this also introduces a form of communication between the two levels. As the lower-level improves, the adjacent region would expand accordingly. What can be missing for applying adjacency constraints in subgoals in dynamic environments? I feel it could be an interesting discussion to add to the paper, highlighting the necessity of timed subgoals and learning timing.

(4)	What is the rationale behind only assigning positive rewards when reaching the subgoal exactly at $\Delta t = 0$ (eqn. (4))? What would happen if the agent reached the goal before using up the timesteps? Any bonus? For example, if $\Delta t = 1$ and the agent reached the subgoal already, one could adjust the data and store $\Delta t^0_t = \Delta t^0_t – 1$ in the buffer, indicating the agent reached the goal sooner than expected. What might be wrong with this design, i.e., rewarding the lower-level if reaching the goal within $\Delta t$ steps?

Minor Comments:

(1)	(line 160): I believe should be $t + \Delta t$ instead of $s + \Delta t$.

(2)	(line 191-192): eqn. (3) $r^1(s_t, a^1_t)$?



----- Update -----

The authors have addressed my questions in their response and provided additional experiments on static benchmark environments. The results look promising, and I have raised my score accordingly.


Ref.:
[1] Zhang, Tianren, Shangqi Guo, Tian Tan, Xiaolin Hu, and Feng Chen. "Generating Adjacency-Constrained Subgoals in Hierarchical Reinforcement Learning." arXiv preprint arXiv:2006.11485 (2020).




**Time Spent Reviewing:**

3.5

---

> ### Author Response · Authors · 2021-08-10
> **Response to 3XHt**
>
> Thank you very much for the detailed and constructive review.
>
> > (1): Section 3.1 could be better organized to improve clarity. I spent most of the time understanding all the algorithmic details in section 3.1. It could be better to break down this section into subsections where each subsection highlights and introduces one aspect of the algorithm.
>
> We agree that a clearer structure would improve the readability of Section 3.1 and we will therefore break it down into subsections. We have also added a more detailed diagram explaining the algorithm (see Fig. +1 on the [Project page](https://sites.google.com/view/hrl-with-timed-subgoals/home)).
>
> > 1a.: (line 215-216) says “the lower level receives a non-zero reward only if it achieves a timed subgoal”. This is inconsistent with the equation above line 97, which assigns zero rewards to the lower-level when near the subgoal.
>
> Section 2.1 is concerned with conventional subgoal-based HRL and the equation above line 97 applies to HAC and not to HiTS. We will emphasize this at the beginning of Section 2.1.
>
> > 1b.: (line 233-235) reads “Furthermore, while pursuing … the higher level is penalized.” I feel like this part needs more elaboration.
>
> We kept our discussion of testing transitions to a minimum as they were already introduced and discussed in [2]. Given the extra page we will elaborate on it more.
>
> > 1c.: (line 237-239) reads “While these two exceptions … respect the limitations of the lower level”. It would be better to clearly mention that this part is the limitation of the proposed algorithm.
>
> We agree and will therefore mention this aspect when discussing limitations in the conclusion.
>
> > 1d.: Is there any transition function used for $g^0$  from $t$ to $t+1$ ? It could be added to improve clarity.
>
> $g^0$ is left unchanged from $t$ to $t+1$, i.e., the transition function is the identity (We use absolute goals, not directional goals). We have made this explicit in the new diagram (Fig. +1 on the [Project page](https://sites.google.com/view/hrl-with-timed-subgoals/home)).
>
> > 2a.: Although discussed in the paper, the high variance of HiTS in the experiments does call for further investigation and analysis. How many random seeds are used for generating the results (I don’t think it is mentioned in the paper)? Does that mean the non-stationarity in high-level learning still exists? Is the performance dependent on choosing the right seed? Can this be alleviated by conservative exploration as hypothesized in the paper?
>
> Details about the training such as the number of random seeds are discussed in Supplementary Material C.2. The results for HiTS presented in the main text are based on 30 random seeds each. The higher level still sees a weak non-stationarity caused by the subgoal achievement tolerance and testing transitions as discussed in line 231 ff. We argue that the high variance can be mostly attributed to the challenging dynamic environments and not to the design of HiTS. This argument is supported by HiTS exhibiting a variance comparable to HAC on less challenging static tasks (Fig. S1 in the Suppl. Material). We briefly mentioned this argument in line 335 ff. and will elaborate on this in the revision.
>
> > 2b.: Since the three dynamic environments are likely to be new to the readers and they all focus on learning the “right timing”, it worth evaluating and comparing HiTS on commonly-used benchmark environments as well. I would suggest bringing the additional static experiments on UR5Reacher and AntFourRooms back to the main paper, which are more familiar by the readers. Can “increasing the subgoal budget” improve the learning efficiency for HiTS on AntFourRooms? It would be interesting to see and compare results with different budgets, and see how well HiTS can perform on static benchmarks.
>
> As mentioned in the abstract and the introduction our algorithm is targeted at dynamic environments. As demonstrated in Fig. S1 HiTS can learn on static environments too. However, we do not expect to outperform subgoal-based methods like HAC in this setting (see paragraph starting with line 35 for a discussion). Due to the restricted page count, we lean towards concentrating on dynamic environments in the main text and keeping the experiments with static environments in the Supplementary Material. In Fig. +3 on the [Project page](https://sites.google.com/view/hrl-with-timed-subgoals/home) we varied the subgoal budget on Platforms while keeping the rest of the hyperparameters fixed.
>
> > (3): To my understanding, timed subgoals highlight the need for communication between high- and lower-levels for concurrent and effective learning. Can other forms of communication (partly) address the non-stationarity issue in practice? Mentioned in (line 338 – 341), generating feasible subgoals seems to be important for learning efficiency. Recently, [1] proposed an algorithm, HRAC, that generates subgoals that are k-step adjacent to the current state, improving the reachability of subgoals at the lower-level. Since the adjacency is learned by sampled trajectory data and updated periodically, this also introduces a form of communication between the two levels. As the lower-level improves, the adjacent region would expand accordingly. What can be missing for applying adjacency constraints in subgoals in dynamic environments? I feel it could be an interesting discussion to add to the paper, highlighting the necessity of timed subgoals and learning timing.
>
> We agree that choosing the right mode of communication between the levels is crucial for sample-efficient learning in HRL. We believe that using an adjacency loss similar to HRAC [1] on the higher level of the HiTS hierarchy instead of using testing transitions could improve sample efficiency in the case of a low-dimensional subgoal space. We thank you for highlighting this possibility. We will discuss this potential direction for future work in the related work section.
>
> > (4): What is the rationale behind only assigning positive rewards when reaching the subgoal exactly at $\Delta t = 0$ (eqn. (4))? What would happen if the agent reached the goal before using up the timesteps? Any bonus? For example, if $\Delta t = 1$ and the agent reached the subgoal already, one could adjust the data and store $\Delta t^0_t = \Delta t^0_t -1$ in the buffer, indicating the agent reached the goal sooner than expected. What might be wrong with this design, i.e., rewarding the lower-level if reaching the goal within steps?
>
> To avoid misunderstandings, let us point out that if control was handed back to the higher level at $\Delta t > 0$ during a rollout, this would introduce a non-stationarity as the transition times in the induced SMDP would change with the learning progress of the lower level. Having said that, we agree with you that relabeling experience on the lower level so as to learn from achieving a goal early is a good idea. It indeed helps with generalizing from achieving a goal early to achieving it at the right time. Our algorithm is actually using relabeled transitions like this as they are a special case of HER adapted to timed subgoals as detailed in equation (6) (where the achieved goal $\hat{g}^0_t$ happens to be the assigned goal $g^0_t$ and $n < \Delta t^0_t$).
>
> Thank you very much for pointing out two typos!
>
> > Limitations And Societal Impact: The authors discussed the limitation but did not address negative social impact. The proposed algorithm helps improve learning in complex and long-horizon dynamic environments. As an illustrative example, what if the algorithm was used by a malicious user to build a robot? Targeting rewards that lead to some harmful behaviors for a society (in a dynamic environment)? I believe such kind of scenario could be discussed. In addition, what can and should be done to prevent it from happening?
>
> The improved performance in dynamic environments could be used to avoid harming humans or animals (by avoiding collisions, assisting the elderly in case of a fall etc.) but could also be used to target them. We will briefly mention the potential for misuse in the conclusion.
>
> References:
>
> [1] Zhang, Tianren, Shangqi Guo, Tian Tan, Xiaolin Hu, and Feng Chen. "Generating Adjacency-Constrained Subgoals in Hierarchical Reinforcement Learning." arXiv preprint arXiv:2006.11485 (2020).
>
> [2] A. Levy, G. Konidaris, R. Platt, and K. Saenko. Learning Multi-Level Hierarchies with
> Hindsight. In International Conference on Learning Representations, 2019.

---

> > ### Comment · Reviewer_3XHt · 2021-08-23
> > **Thank you**
> >
> > Thanks very much for the detailed response. I believe most of my questions have been clarified.
> >
> > A common concern is the comparison and performance on static benchmark environments. I believe it would be very helpful to show that HiTS is a general and effective approach for solving general/any HRL problems. (a) Can the performance of HiTS on AntFourRooms be improved by adjusting/fine-tuning the subgoal budget? (which was also mentioned in the supplementary material) (b) Would it be possible to show that HiTS can achieve similar or better performance against baselines on some other static environments included in [1]? I wonder if this could better demonstrate the effectiveness and generality of HiTS on static benchmarks.
> >
> > [1] Levy, Andrew, et al. "Learning multi-level hierarchies with hindsight." arXiv preprint arXiv:1712.00948 (2017).

---

> > > ### Author Response · Authors · 2021-09-02
> > > **Additional experiments on standard benchmarks**
> > >
> > > We thank you for your feedback and apologize for the late response. We used the time for additional experiments on standard benchmark tasks which we also discuss in a general comment above.
> > >
> > > > (a) Can the performance of HiTS on AntFourRooms be improved by adjusting/fine-tuning the subgoal budget? (which was also mentioned in the supplementary material)
> > >
> > > We included the subgoal budget and separate learning rates for the two levels in the hyperparameter optimization on AntFourRooms (see Fig. +6 on the [Project page](https://sites.google.com/view/hrl-with-timed-subgoals/home) ). The performance of HiTS did indeed improve but so did HAC. As a result, HAC still learns faster than HiTS in this environment even though the performance of HiTS is already quite good, in fact better than what was reported for the original implementation of HAC in [1].
> > >
> > > > (b) Would it be possible to show that HiTS can achieve similar or better performance against baselines on some other static environments included in [1]?
> > >
> > > We also compared HiTS to HAC in the Pendulum environment from [1] (see Fig. +5 and Fig. +6 on the [Project page](https://sites.google.com/view/hrl-with-timed-subgoals/home)). As already observed for UR5Reacher, HiTS outperforms HAC on this task. We hypothesize that this is due to the lower level being able to generalize over $\Delta t$ whereas HAC has to learn how to do movements faster from trial and error.
> > >
> > > We additionally considered the Ball in cup environment from [2]. Even though the task is rather simple, HiTS slightly outperforms the strong SAC baseline while HAC suffers from a big variance and lower asymptotic performance, as expected in a dynamic environment.
> > >
> > > As HiTS outperforms HAC on three of the four considered standard benchmark tasks, performed well on AntFourRooms and is able to learn on the challenging dynamic tasks we proposed, we consider our algorithm to be generally applicable, not only to dynamic but also to static environments. We will include the additional results in the main text.
> > >
> > > [1] Levy, Andrew, et al. "Learning multi-level hierarchies with hindsight." arXiv preprint arXiv:1712.00948 (2017).
> > >
> > > [2] Tassa et al, 2020. dm_control: Software and Tasks for Continuous Control, arXiv 2006.12983.

---

> > > > ### Comment · Reviewer_3XHt · 2021-09-03
> > > > **Re: Additional experiments**
> > > >
> > > > Thanks very much for the additional results. My concerns have been addressed and I have raised my score.

---

### Official Review · Reviewer_z3aR · 2021-07-18

**Rating:** 7
**Confidence:** 4

**Summary:**

The paper proposes to study HRL with timed goals wherein the high level controller can not only specify subgoals but also the time when it should be reached. The overarching goal of the work is to enable HRL methods to adapt better to non-stationarity. The key contributions of this work are 1) propose the idea of timed subgoals to address the adverse impact of non-stationarity and ever changing low level behaviors on the overall policy, 2) distil these insights into an HRL algorithm and introduce HiTS as the proposed algorithm, 3) introduce tasks where such challenges are seen, and 4) the empirical analysis shows sample-efficient learning which results in stable agents as opposed to existing methods which fail. Finally, the authors also show theoretically the utility of the timed-subgoals in eliminating non-stationarity in the data generating process for the high level controller.

**Ethical Concerns:**

None.

**Limitations And Societal Impact:**

While Sec 3.1 and Sec 4.1 cover limitations, it is hard to find at a first look where limitations are discussed in rigorous detail. I would suggest adding a paragraph in the conclusion section.

I do not see a broader impact statement, I encourage you to write one and add that to the paper.


**Main Review:**

**Strengths** :
This work tackles a very interesting problem on how HRL algorithms should be designed for changing behaviors at lower levels during training. The dynamic environments bring this problem to a forefront as the low level policies continue to adapt - thus resulting in the high level controller unable to learn effectively either as the transition times might vary. The paper presents new creative ideas to address such challenges through the notion of time-subgoals. While existing methods such as hindsight action relabeling techniques can alleviate this problem for static environments, this work shows that for dynamic environments, action relabeling does not prevent different environment outcomes for slow and fast low-level policies. This is accomplished by restricting the subgoal space to the directly controllable part of the state (sc) , and conditioning the lower level on a timed subgoal, replacing the external component of the state with time.

**Weaknesses**:

* This approach requires the assumption that the environment dynamics are independent from the agent’s state - many situations do not warrant this assumption.
* Moreover an additional assumption made is that when replacing the action via hindsight, action relabeling completely removes the non-stationarity of the SMDP the level is interacting with. Would it be possible to relax the first assumption in this framework as it stands?


**Empirical Analysis**:
The experiments test HiTS sample efficiency and stability on three different tasks including platforms, drawbridge, and tennis 2D. All of these tasks require the agent to be precise in the timing of certain actions. The experiments in Fig 4 show that the proposed algorithm is able to outperform the baselines and give evidence for 1) aided exploration via temporal abstraction and 2) Stability induced via timed subgoals. However, I have the following concerns and questions:


* Re the claim that ”higher level explicit control over the degree of temporal abstraction, giving it more direct access to the tradeoff between a small effective problem horizon and exercising tight control over the agent.” How do you show this in your experiments? Is there an analysis that can support this concretely?  In particular, the phrase tight control over the agent is very loosely used unless I am missing something. It would be useful to rephrase this.

* Re the claim that “The explicit representation of timing greatly facilitates credit assignment in these cases.”  To support this claim, Fig. 5 shows how the discrepancy between the moment of contact between ball and racket and the closest achievement time of a timed subgoal- but it is not very clear to me how this demonstrates the credit is assigned correctly? Can you provide additional analysis to support this claim? It would add value to the paper, if you can show a plot with the ground truth of when a robot state should be achieved vs the one determined by the proposed approach?

* Re “interestingly, augmenting the state, observation and goal spaces with absolute time does not improve the performance of the two-level HAC hierarchy but impedes it instead.” This seems a bit counterintuitive to me atleast. Do you have any insights on why this would hurt HAC as opposed to helping it?
For the drawbridge experiment, what happens if we increase the subgoal budget? Does that help HAC perform better?


**Writing and Presentation**: The paper is very well written and ideas are clearly explained. I find the presentation of the main ideas, especially motivation very clear.



**Questions and Misc.**:

* Re “This definition of the hierarchy fixes the transition time the higher level sees \tau” Is my understanding correct that this would result in fixed subgoal lengths or that each subgoal can still be of variable lengths?
* Re “ The higher level can then learn to identify the situations in which it has to exercise tight control over the agent because it is about to influence the environment and align its choice of desired subgoal achievement times to them.” How is this done at the higher level precisely? While the details in the Sec 3.1 are clear, it would be great if you can expand on the intuition behind how does the higher level really model the dynamic component of the environment, as that is what translates to specifying the timed-subgoals.


**Time Spent Reviewing:**

5

---

> ### Author Response · Authors · 2021-08-10
> **Response to z3aR**
>
> Thank you for your constructive and detailed review.
>
> > This approach requires the assumption that the environment dynamics are independent from the agent’s state - many situations do not warrant this assumption.
>
> > Moreover an additional assumption made is that when replacing the action via hindsight, action relabeling completely removes the non-stationarity of the SMDP the level is interacting with. Would it be possible to relax the first assumption in this framework as it stands?
>
> It is indeed possible to relax the assumption that the agent’s state does not influence the environment to an interaction that is sparse in time as discussed in line 170 ff. If the higher level aligns the subgoal achievement times with these interactions, using hindsight action relabeling will still remove the non-stationarity of the SMDP. In practice, the alignment does not have to be perfect for sample-efficient learning to be possible as depicted in Fig. 5.
>
> > Re the claim that ”higher level explicit control over the degree of temporal abstraction, giving it more direct access to the tradeoff between a small effective problem horizon and exercising tight control over the agent.” How do you show this in your experiments? Is there an analysis that can support this concretely? In particular, the phrase tight control over the agent is very loosely used unless I am missing something. It would be useful to rephrase this.
>
> In order to substantiate this statement, we added a figure (Fig. 2+ on the [Project page](https://sites.google.com/view/hrl-with-timed-subgoals/home)) that shows how HiTS and HAC distribute their subgoal budget over episodes of the Tennis2D environment. The x-axis shows the time relative to the contact between racket and ball. While HiTS uses more subgoals around the time of the contact and less during the rest of the episode, HAC distributes its subgoals more uniformly. This behavior can also be seen in the video. We argue that the higher level in HiTS can explicitly choose small time intervals $\Delta t$ around the time of the contact to stay reactive (i.e., exercise “tight control” over the agent) while choosing bigger $\Delta t$ during the rest of the episode when the agent can afford temporal abstraction. Hence, HiTS can adapt its (non-uniform) temporal abstraction to the task. We will add a discussion of this figure to the experiments section.
>
> > Re the claim that “The explicit representation of timing greatly facilitates credit assignment in these cases.” To support this claim, Fig. 5 shows how the discrepancy between the moment of contact between ball and racket and the closest achievement time of a timed subgoal- but it is not very clear to me how this demonstrates the credit is assigned correctly? Can you provide additional analysis to support this claim? It would add value to the paper if you can show a plot with the ground truth of when a robot state should be achieved vs the one determined by the proposed approach?
>
> Intuitively, the argument is that if the timing is crucial for the task then representing it explicitly in the action of the higher level should make it easier to assign credit to it. In particular, because the reduced non-stationarity of the induced SMDP makes it easier to learn precise timing early in the training without having to readjust it due to the changing lower level. A ground-truth timing is not easy to obtain. We are working on a good visualization, however, for the moment we will remove this claim from the introduction, also as it has an overlap with the improved stability due to timed subgoals.
>
> > Re “interestingly, augmenting the state, observation and goal spaces with absolute time does not improve the performance of the two-level HAC hierarchy but impedes it instead.” This seems a bit counterintuitive to me at least. Do you have any insights on why this would hurt HAC as opposed to helping it? For the drawbridge experiment, what happens if we increase the subgoal budget? Does that help HAC perform better?
>
> The HAC baseline with an observation and subgoal space that has been augmented with time has been briefly discussed in lines 177 ff. and 274 ff. and the reasons for its poor performance have been mentioned in 292 ff. We will dedicate more space to a discussion of this baseline in the experiments section and will add roughly the following material:
> The observation in this baseline is given by the state $s$ and the time $t$ that has passed in the current episode, $ (s, t) \in \mathcal{S} \times \mathbb{N}_{\geq0} $ .
>
> An augmented subgoal $(g, \bar{t})\in \mathcal{G}\times \mathbb{N}_{\geq0}$ is achieved if the state $s^c$ of the agent is sufficiently close to $g$ and the current time $t$ is close to the desired achievement time $\bar{t}$. The subgoals used by this baseline hierarchy consequently contain information about when they are to be achieved but the HAC algorithm is left unchanged.
>
> In contrast to HiTS, this baseline (i) communicates in terms of absolute time (ii) does not automatically query a new subgoal at $t=\bar{t}$ if the current one was not achieved and (iii) uses the HAC reward (eqn. above line 97) instead of eqn. (4) on the lower level. This causes several problems, most notably:
> 1. conditioning the lower level on absolute time introduces a spurious time dependence into the policy
> 2. the agent gets “stuck” on a missed subgoal until the action budget of the lower level is exhausted. The second point, in particular, is a limiting factor as this happens frequently in an episode and completely “paralyzes” the agent.
>
> We have added a figure (Fig. +3 on the [Project page](https://sites.google.com/view/hrl-with-timed-subgoals/home)) that shows the performance of HiTS and HAC on the Platforms environment as a function of the subgoal budget (while keeping other hyperparameters fixed). It looks like HAC profits a bit from increasing the number of subgoals but it does not improve significantly.
>
> > Re “This definition of the hierarchy fixes the transition time the higher level sees $\tau$” Is my understanding correct that this would result in fixed subgoal lengths or that each subgoal can still be of variable lengths?
>
> You are correct in that the time interval during which the lower level will pursue a given subgoal is fixed by the action of the higher level (containing $\Delta t$). This $\Delta t$ can be freely chosen by the higher level, however. In a typical episode, HiTS will choose varying $\Delta t$ (this can be seen in the part of the video dedicated to the Tennis2D environment, for example).
>
> > Re “The higher level can then learn to identify the situations in which it has to exercise tight control over the agent because it is about to influence the environment and align its choice of desired subgoal achievement times to them.” How is this done at the higher level precisely? While the details in the Sec 3.1 are clear, it would be great if you can expand on the intuition behind how does the higher level really model the dynamic component of the environment, as that is what translates to specifying the timed-subgoals.
>
> In the current version of HiTS, the higher level does not explicitly learn a model of the environment dynamics. It is still incentivized to align the subgoal achievement times with those points in time when the controllable part interacts with the rest of the environment since this helps with maximizing the return (e.g. in the Tennis2D environment): The lower level, which is trained using a maximum entropy objective, is particularly stochastic in between timed subgoals as it can afford to explore there. When $\Delta t$ is close to $0$, on the other hand, the noise on the agent’s state is small since this is necessary for achieving the timed subgoal (see Fig. +4 on the [Project page](https://sites.google.com/view/hrl-with-timed-subgoals/home)). Hence, the higher level is encouraged to align the subgoal achievement times with those points in time when agent and environment interact so as to solve the task unimpeded by low-level noise. This mechanism is discussed in Supplementary Material B.2. It is an interesting direction for future work that could improve sample efficiency to more actively guide the alignment of timed subgoals with interactions between agent and environment.
>
> > While Sec 3.1 and Sec 4.1 cover limitations, it is hard to find at a first look where limitations are discussed in rigorous detail. I would suggest adding a paragraph in the conclusion section.
> I do not see a broader impact statement, I encourage you to write one and add that to the paper.
>
> We will summarize limitations and briefly discuss societal impact in the conclusion.

---

> > ### Comment · Reviewer_z3aR · 2021-08-20
> > **Thanks for your response, probing on significance of this work.**
> >
> > Thank you for a thorough response to my concerns and criticism. I find that most of the questions and concerns I had have been addressed in the rebuttal response by the authors.
> >
> > A common concern for multiple reviewers is the **comparison on standard benchmarks** and therefore the question on the **significance of this work**. As I understand in the rebuttal response in general, the argument authors make is that the method is not expected to outperform other HRL methods in static environments.
> >
> > One way to demonstrate the motivation and significance of the proposed approach is to demonstrate the shortcomings of existing baselines in the proposed dynamic environments. What is the core reason to chose HAC as the main HRL baseline? Do you anticipate the conclusions generalise to other HRL baselines? If so, can you pinpoint what aspects of the solutions found by HITS will guarantee this intuitively? I wonder if this exercise will help further clarify the essence of the proposed work.

---

> > > ### Author Response · Authors · 2021-08-24
> > > **On the scope of HiTS and a lack of benchmarks in dynamic environments**
> > >
> > > Thank you for pointing out where our argument could benefit from further clarifications.
> > >
> > > > A common concern for multiple reviewers is the comparison on standard benchmarks [...]
> > >
> > > We agree that standard benchmarks are important as a point of reference. However, we are not aware of any well-known challenging control benchmarks in dynamic environments. In particular, the subgoal-based HRL literature is quite focused on static maze-like environments (if objects are present in the environment, then they immediately come to rest when not pushed by the agent as in [3]). We therefore saw the need to introduce new benchmark tasks.
> > >
> > > > As I understand in the rebuttal response in general, the argument authors make is that the method is not expected to outperform other HRL methods in static environments.
> > >
> > > Our motivation for developing HiTS was indeed to improve sample efficiency on tasks in dynamic environments. While options can be used in dynamic environments, learning them in parallel with a policy over options is not necessarily more sample-efficient than a non-hierarchical baseline [1]. In contrast to this, subgoal-based HRL methods clearly outperform flat baselines in static environments [2,3,4]. Thus, the question whether the improved sample efficiency of subgoal-based HRL methods can be transferred to dynamic environments arises naturally. As dynamic elements like a conveyor belt, cars, humans, animals etc. are ubiquitous in real-world applications, we believe that developing algorithms that can learn efficiently in their presence is a significant research area.
> > >
> > > > What is the core reason to chose HAC as the main HRL baseline?
> > >
> > > Our main reason for choosing HAC as a baseline was its sample efficiency. In a direct comparison [4] HAC outperformed HIRO [3] on a variety of environments, probably due to its use of hindsight. Furthermore, HIRO was shown to be more sample-efficient than FuN [2] and SNN4HRL [5] in [3]. Moreover, unlike FuN and SNN4HRL, the design of HAC partly addresses the non-stationarity issue that arises when training multiple levels in parallel. We therefore expect HAC to be the hardest subgoal-based HRL baseline to beat.
> > >
> > > > Do you anticipate the conclusions generalise to other HRL baselines?
> > >
> > > We do expect other conventional subgoal-based HRL methods to struggle just as much - if not more - than HAC on dynamic environments. SNN4HRL pretrains low-level skills, freezes them and lets a high-level policy activate them for a fixed number of time steps. As the skills are not adapted to the task, the higher level has no control over timing and interactions with the environment and can be expected to fail on a dynamic downstream task. HIRO and FuN reward the lower level for being close to and progressing towards the subgoal, respectively. Because the low-level policy will get faster during training, the environment state resulting from assigning one and the same subgoal will change as well (see Section 2.2). Hence, the same non-stationarity issue that plagues HAC in dynamic environments will occur with HIRO and FuN as well. In the paper we focused our discussion on HAC mostly for concreteness.
> > >
> > > > If so, can you pinpoint what aspects of the solutions found by HITS will guarantee this intuitively? I wonder if this exercise will help further clarify the essence of the proposed work.
> > >
> > > We motivated the design of HiTS in the beginning of Section 3 and in the introduction (line 60 ff.). We furthermore formalized the essence of the algorithm in proposition 1 and 2 but we agree that an intuitive explanation of why HiTS works should be added after the algorithm has been introduced. We will therefore add a paragraph to the final version along the lines of:
> > > Intuitively, what rids HiTS of the non-stationarity is that from the perspective of the higher level assigning a feasible timed subgoal to the lower level will always have roughly the same effect in a given situation. The objective of the lower level is to be “at the right place at the right time”. Once it has learned how to do that it might become slightly more precise in fulfilling that task but it won’t try to arrive “early”. Hence, invoking the lower level has a reliable effect on the agent and the environment and the higher level can start to learn a stable solution already early on.
> > >
> > > We are of course happy to respond to additional questions or comments.
> > >
> > > [1] P.-L. Bacon, J. Harb, and D. Precup. The option-critic architecture. In Proceedings of the AAAI Conference on Artificial Intelligence, volume 31, 2017.
> > >
> > > [2] A. S. Vezhnevets, S. Osindero, T. Schaul, N. Heess, M. Jaderberg, D. Silver, and K. Kavukcuoglu. FeUdal networks for hierarchical reinforcement learning. volume 70 of Proceedings of Machine Learning Research, pages 3540–3549, International Convention Centre, Sydney, Australia,
> > > 06–11 Aug 2017. PMLR.
> > >
> > > [3] O. Nachum, S. S. Gu, H. Lee, and S. Levine. Data-efficient hierarchical reinforcement learning. In Advances in Neural Information Processing Systems, pages 3303–3313, 2018.
> > >
> > > [4] A. Levy, G. Konidaris, R. Platt, and K. Saenko. Learning Multi-Level Hierarchies with
> > > Hindsight. In International Conference on Learning Representations, 2019.
> > >
> > > [5] Carlos Florensa, Yan Duan, and Pieter Abbeel. Stochastic neural networks for hierarchical
> > > reinforcement learning. arXiv preprint arXiv:1704.03012, 2017.

---

> > > > ### Author Response · Authors · 2021-09-02
> > > > **Additional experiments on standard benchmarks**
> > > >
> > > > We would like to add that we performed additional experiments on standard benchmarks in order to help decide the question of the significance of our contribution. We discuss these results in a general comment that can be found above and we will add them to the main text.
> > > >
> > > > We thank you for your feedback!

---

### Official Review · Reviewer_uHJN · 2021-08-16

**Rating:** 5
**Confidence:** 3

**Summary:**

This paper proposes a new method for HRL in which the higher level chooses both a subgoal and a time at which the subgoal should be achieved ("Hierarchical RL with Timed Subgoals, or HITS). The main claim of the work is stated early on: "Hence, the use of timed subgoals extends the class of tasks that can be solved efficiently by subgoal-based HRL methods".

This claim is motivated by several elements. First, the paper identifies an issue of non-stationarity in dynamic environments: the higher level's reliance on the lower level's behavior induces a particular kind of non-stationarity; as learning progresses, the lower level behaviors will improve, thereby changing the outcome of the same high level actions. This discrepency motivates the claim that "all levels in a hierarchy should see transitions that look like they were generated by interacting with a stationary effective environment." Alongside this, it is noted that dynamic environments with elements not controlled by the agent are particularly challenging as they further introduce non-stationarity. Second, the paper focuses on a decomposition of state into the controlled and uncontrolled parts of state, which allows for subgoals to concentrate on the controlled aspects of the environment. These two points are examined to motivate timed subgoals alongside hindsight relabeling techniques to form the core of HITS. Otherwise, by my reading, HITS is quite simple (which I take to be a virtue). Three new benchmark domains are introduced that feature time-based reasoning, such as a ship that needs to carefully furl its sail in order to quickly pass through a drawbridge. These domains are chosen to highlight the positive aspects of HITS, and to differentiate what HITS does differently from other HRL methods. The results quite clearly demonstrate that HITS is well suited to solve these domains.

**Main Review:**

The paper reads well, and to my knowledge the specific proposal of including a completion-time in the high level is new. The domains introduced showcase the strengths of HITS, but I also worry that they are too focused on the specific characteristics that HITS is developed for. In other words, I believe it would be a useful empirical test to also showcase HITS on more traditional domains to examine the impact that the added timed-subgoals have in general. Do they lead to degenerate solutions? This suggestion feeds into my overall assessment of the paper: the main claim of the work is that timed subgoals can help HRL. I believe the chosen domains and experiments demonstrate that timed subgoals help when certain well chosen properties are present. After reading, I am unclear how big of a problem these non-stationariy issues are in general, and it is unclear how general timed subgoals are as a solution. In more detail:

**originality** As stated, to my knowledge this particular combination is novel. Naturally the paper borrows a lot from Levy et al. 2018, but it is transparent about this fact. There is perhaps some overlap with work on Linear Temporal Logic and Reward Machines in RL, which both focus on learning to solve tasks that involve particular, possibly complex, temporal sequences. See Camacho et al. 2017 ("LTL and Beyond") or Littman et al. 2017 ("Environment-Independent Task Specifications via GLTL"). I don't believe these need to be cited or discussed in the work, but the insights there might be useful.

**clarity** I found the paper to be relatively clear, though I do believe that the stated propositions are stated too abstractly. I appreciate not getting bogged down by details but it is difficult to infer the precise meaning of these statements without further formalism. For instance, what does "consistent" in Prop. 2 mean? What is the nature of the stationary SMDP? Separately, I believe the _precise_ nature of the "problem" described in the introduction could be formalised and stated more precisely. Several aspects are mentioned: That the environment contains dynamic elements no controlled by the agent and that the learning dynamics in high & low level induce a particular kind of non-stationarity. I believe honing in on the precise problem would benefit the work. It is hard to tell how general this problem is, as I found the essence of the problem to still be unclear.

**significance** As stated above, my main reaction is that it is unclear how significant the proposed problem is, and thus, how useful or general the proposed solution is. For this reason my assessment is that the significance of the work is on the lower side.


**Time Spent Reviewing:**

1

---

> ### Comment · Area_Chair_JbAi · 2021-08-16
> **Additional review**
>
> To avoid any confusion on the part of authors and/or reviewers: this is an additional review solicited as the paper only had 3 (instead of 4) reviews which also seemed to put the paper close to the borderline. I invite the authors to react to this additional review and the other reviewers to consider the points in this review in the discussion. Note that the timeline for an emergency review is quite short, so this review might be less exhaustive than average.

---

> ### Author Response · Authors · 2021-08-24
> **Response to review**
>
> Thank you for your comments on our submission and for taking time at short notice. We are happy that you share our opinion on the novelty and soundness of our work.
>
> > Separately, I believe the precise nature of the "problem" described in the introduction could be formalised and stated more precisely. Several aspects are mentioned: That the environment contains dynamic elements no controlled by the agent and that the learning dynamics in high & low level induce a particular kind of non-stationarity. I believe honing in on the precise problem would benefit the work. It is hard to tell how general this problem is, as I found the essence of the problem to still be unclear.
> > After reading, I am unclear how big of a problem these non-stationary issues are in general, and it is unclear how general timed subgoals are as a solution.
> > As stated above, my main reaction is that it is unclear how significant the proposed problem is, and thus, how useful or general the proposed solution is. For this reason my assessment is that the significance of the work is on the lower side.
>
> We dedicated Section 2.2 and in particular line 130 ff. to a detailed discussion of the non-stationarity problem in dynamic environments which we outlined in the introduction. In essence, the distribution of the state $s^e$ of the not directly controllable part of the environment that is reached when achieving a conventional subgoal in $\mathcal{G}=S^c$ depends on the low-level policy even if hindsight action relabeling as introduced in [1] is used (see line 141 ff.). If the not directly controllable part of the environment is relevant to solving the task this means that the higher level cannot start to learn efficiently until the lower level is converged.
>
> This issue will arise in **any** task which involves not only the agent but also other dynamic elements like a conveyor belt, cars, humans, animals etc. It is therefore quite general and can be expected to hamper the application of subgoal-based methods to real-world problems (see discussion in line 56 ff.). In line with our theoretical analysis, our experiments on relatively simple dynamic environments illustrate that this non-stationarity issue prevents a conventional subgoal-based HRL algorithm from learning a stable solution.
>
> As stated in Proposition 2 and proved in Section A of the Supplementary Material, timed subgoals in conjunction with hindsight action relabeling **are a general solution** to the non-stationarity issue in dynamic environments as they completely remove the non-stationarity of the induced SMDP that generates the data the higher level learns from.
>
> > The domains introduced showcase the strengths of HITS, but I also worry that they are too focused on the specific characteristics that HITS is developed for. In other words, I believe it would be a useful empirical test to also showcase HITS on more traditional domains to examine the impact that the added timed-subgoals have in general.
> > I believe the chosen domains and experiments demonstrate that timed subgoals help when certain well chosen properties are present.
>
> As mentioned in the abstract and the introduction the design of HiTS is targeted at dynamic environments. The proposed benchmark tasks were constructed to highlight the challenges that a learning agent faces in such environments, i.e., precise timing, waiting and adapting its trajectory to the dynamic elements beyond its control. Other than involving a dynamic environment with which the agent interacts sparsely, the tasks are rather generic.
> However, we agree that standard benchmarks are important as a point of reference. Unfortunately we are not aware of any well-known challenging control benchmarks in dynamic environments. We therefore saw the need to introduce our own environments.
>
> In terms of traditional experiments, we included results on two static environments introduced in [1]. The results are presented in Fig. S1 in the Supplementary Material and illustrate that HiTS reaches an asymptotic performance similar to that of HAC. However, we do not expect to outperform subgoal-based methods like HAC in terms of sample efficiency in this setting. The reason is that for navigation tasks in a static environment the non-stationarity of the induced SMDP is in line with the task (see paragraph starting with line 35 for a discussion).
>
> The motivation for our work was to break out of the “subgoal-based HRL is for static, maze-like environments” paradigm and to demonstrate that gains in sample efficiency can be transferred to challenging dynamic environments by using timed subgoals.
>
> > Do they [timed subgoals] lead to degenerate solutions?
>
> We would like to ask you to elaborate on this question as the meaning of degenerate is not clear to us in this context. If you are referring to sequences of timed subgoals with time intervals $\Delta t = 1$ then the answer is no, we do not observe such solutions at the end of the training as the timed subgoal budget is limited and the higher level is penalized for every emitted timed subgoal as specified in equation (3).
>
> > I found the paper to be relatively clear, though I do believe that the stated propositions are stated too abstractly. I appreciate not getting bogged down by details but it is difficult to infer the precise meaning of these statements without further formalism. For instance, what does "consistent" in Prop. 2 mean? What is the nature of the stationary SMDP?
>
> We chose high-level formulations for the propositions so as to comply with the restricted page count. We agree with you that being more explicit would improve clarity. We will therefore add the following explanation to Proposition 2:
> More precisely, the distribution of high-level transitions in the replay buffer can be decomposed as $p_t\left(s, a, \tau, s’, r\right)=p_t\left(s, a\right)p\left(s’, \tau, r \mid s, a\right)$ where the conditional probability $p\left(s’, \tau, r \mid s, a\right)$ corresponding to the induced SMDP is time-independent [$\tau$ denotes the transition time].
>
> For a discussion of this conditional probability and how the use of timed subgoals with hindsight action relabeling renders it time-independent we refer to the proof of Proposition 2 (line 539 ff. in the Supplementary Material).
>
> > There is perhaps some overlap with work on Linear Temporal Logic and Reward Machines in RL, which both focus on learning to solve tasks that involve particular, possibly complex, temporal sequences. See Camacho et al. 2017 ("LTL and Beyond") or Littman et al. 2017 ("Environment-Independent Task Specifications via GLTL").
>
> Thank you for mentioning these works. We believe our contribution is largely orthogonal to the cited papers. While Camacho et al. and Littman et al. replace the reward function with a task-specification based on formal languages, we do not assume additional structure, i.e., we learn directly from a sparse reward.
>
> Of course, we are happy to elaborate or answer additional questions.
>
> [1] A. Levy, G. Konidaris, R. Platt, and K. Saenko. Learning Multi-Level Hierarchies with
> Hindsight. In International Conference on Learning Representations, 2019.

---

> > ### Comment · Reviewer_uHJN · 2021-08-25
> > **Re: Response to review**
> >
> > I thank the authors for their thorough and thoughtful response!
> >
> > > We dedicated Section 2.2 and in particular line 130 ff. to a detailed discussion of the non-stationarity problem in dynamic environments which we outlined in the introduction...
> >
> > Understood. I do believe the paper will be sharper with a more precise explanation of the problem. For instance, after reading the paragraph at line 130, I still do not have a crisp understanding of what is meant by "dynamic elements which are not directly controllable by the agent". A few questions still come to mind as I read this passage: Might this also include any stochasticity in the environment? Or, can these elements be deterministic? Does it have to be the case that for _every_ state and action there is no way the agent can alter the outcome of a particular element in order for it to be not directly controllable, or might it be the case that there are just scarce opportunities to impact such elements? Does "direct" controllability imply that the agent might have indirect methods for influencing these elements? If so, what counts as a direct effect, and what is indirect? (For example, in the conveyor belt case, what if the agent can slightly alter the speed of the belt, or its friction coefficient?) Perhaps not all of these questions need a crisp answer after the problem is introduced, but I do believe that the central role the problem plays in the paper should give the problem definition more air time. Can the conditions be stated precisely?
> >
> > > In terms of traditional experiments, we included results on two static environments introduced in [1]. The results are presented in Fig. S1 in the Supplementary Material and illustrate that HiTS reaches an asymptotic performance similar to that of HAC. However, we do not expect to outperform subgoal-based methods like HAC in terms of sample efficiency in this setting. The reason is that for navigation tasks in a static environment the non-stationarity of the induced SMDP is in line with the task (see paragraph starting with line 35 for a discussion).
> >
> > This is precisely what I was hoping for, thank you for pointing this out. I have looked at Figure S1 and believes this gives initial confirmation that HiTS may slightly impact performance, but not by a considerable amount. My concern was that the incorporation of timed subgoals might leave the approach as entirely ineffective in a non-dynamic environment. To connect to your question here:
> >
> > > We would like to ask you to elaborate on this question as the meaning of degenerate is not clear to us in this context.
> >
> > I had intended to ask whether HiTS deteriorates performance on standard (non-dynamic) environments. Some loss in performance might be expected, but if the addition of timed subgoals left an existing (successful) HRL approach incapable of solving standard HRL tasks, this would be an undesirable outcome. I do believe the results in Figure S1 are critical to the story of the approach. (As an aside, I will provide one "standard reviewer response" that slightly more domains of the kind presented in Figure S1 will strengthen the story). I further suggest moving the results from S1 to the main paper. I will think more the reply and follow up with any further thoughts.

---

> > > ### Author Response · Authors · 2021-09-02
> > > **Clarification of “directly controllable” and additional experiments**
> > >
> > > We thank you for your feedback and apologize for the late response. We have used the time to address the question of how general HiTS is by running additional experiments on standard benchmarks.
> > >
> > > >Understood. I do believe the paper will be sharper with a more precise explanation of the problem. For instance, after reading the paragraph at line 130, I still do not have a crisp understanding of what is meant by "dynamic elements which are not directly controllable by the agent". A few questions still come to mind as I read this passage: Might this also include any stochasticity in the environment? Or, can these elements be deterministic? Does it have to be the case that for every state and action there is no way the agent can alter the outcome of a particular element in order for it to be not directly controllable, or might it be the case that there are just scarce opportunities to impact such elements? Does "direct" controllability imply that the agent might have indirect methods for influencing these elements? If so, what counts as a direct effect, and what is indirect? (For example, in the conveyor belt case, what if the agent can slightly alter the speed of the belt, or its friction coefficient?) Perhaps not all of these questions need a crisp answer after the problem is introduced, but I do believe that the central role the problem plays in the paper should give the problem definition more air time. Can the conditions be stated precisely?
> > >
> > > In line 130 ff. we indeed state the problem on an intuitive level using examples and illustrations. We formulate our assumptions more precisely in Proposition 2 (line 165 ff.) and in the following paragraph (line 169 ff.). We begin by assuming **no influence** of the controllable part on the not directly controllable part of the environment, i.e., $p\left({s'}^e\mid s^e) = p({s}'^e\mid s^e, s^c, a\right)$. We then argue that this assumption can be relaxed to an **interaction which is sparse in time** in the sense that $s^c$ influences $s^e$ only in a small number of time steps such that the higher level can align the desired achievement times with these points in time (which happens in practice as illustrated by Fig. 5). We understand that, as a result of this strategy, the problem definition is “spread out” throughout the paper. We will therefore add a figure illustrating these assumptions to Section 2 of the main text (see Figure +7 on the [Project page](https://sites.google.com/view/hrl-with-timed-subgoals/home) ).
> > >
> > > Intuitively, these assumptions ensure that the higher level can control the interaction with the environment by determining the state $s^c$ at the time of the interaction. Note that any subgoal-based hierarchy necessitates such assumptions because otherwise the way the lower level interacts with the environment is “undefined”. This issue is not usually discussed in the literature as the considered environments are either static or their elements have trivial dynamics (they only move when pushed by the agent).
> > >
> > > Returning to the conveyor belt example, it would be fine for the agent to alter the speed now and then by pushing a button. If it adjusted the speed in every timestep, however, the hierarchy may fail to solve the task. In the latter case it would be more appropriate to see the conveyor belt as a part of the agent/the directly controllable part of the environment.
> > >
> > > > I had intended to ask whether HiTS deteriorates performance on standard (non-dynamic) environments. Some loss in performance might be expected, but if the addition of timed subgoals left an existing (successful) HRL approach incapable of solving standard HRL tasks, this would be an undesirable outcome. I do believe the results in Figure S1 are critical to the story of the approach. (As an aside, I will provide one "standard reviewer response" that slightly more domains of the kind presented in Figure S1 will strengthen the story). I further suggest moving the results from S1 to the main paper. I will think more the reply and follow up with any further thoughts.
> > >
> > > We performed additional experiments on standard benchmark tasks (see general comment above) to address your concern. HiTS can solve all tasks and outperforms HAC on three out of four (see Fig. +6 on the [Project page](https://sites.google.com/view/hrl-with-timed-subgoals/home) ). As HiTS is also capable of solving the challenging dynamic tasks we proposed, we think that our claim that it extends subgoal-based HRL to dynamic environments is justified. We will include these results in the main text, thank you for this suggestion.
> > >
> > > Ball in cup [1], one of the benchmark tasks we considered, nicely illustrates that the sparse interaction assumption can be violated in practice (the ball is attached to the agent, i.e., the cup, with a string) and HiTS may still learn efficiently. As most control tasks involve second-order dynamics, it is mostly sufficient to enable the higher level to choose small $\Delta t$ for it to be able to control a continuous interaction between agent and environment.
> > >
> > > [1] Tassa et al, 2020. dm_control: Software and Tasks for Continuous Control, arXiv 2006.12983.

---

### Author Response · Authors · 2021-08-10
**General response**

We thank the reviewers for their valuable feedback. We are encouraged by the assessment of our contribution as a “new and interesting idea” (R 3XHt). Furthermore, we are pleased that the problem of designing hierarchical algorithms for stable learning in dynamic environments is perceived as “very interesting” (R z3aR) and that our ideas on how to tackle it were seen as “creative” (R z3aR). We are glad that the reviewers found the paper to be “well written” (R 3XHt, z3aR, Auot) and “well-positioned with respect to the literature” (R Auot).

We addressed the reviewers’ questions and concerns by performing additional experiments, by providing new figures, and by additional explanations that will be added to the paper. We respond to the reviewers’ questions in comments to their reviews.

We believe that the paper and in particular its presentation and the analysis of our experimental results have greatly benefited from the reviewers’ insightful feedback. We welcome further comments and are happy to elaborate if additional questions arise.

---

### Author Response · Authors · 2021-09-02
**Additional experiments on standard benchmarks**

We thank the reviewers for their valuable feedback. We agree that experiments on standard benchmarks are important for proving the significance of our work. We have therefore used the remaining time of the discussion phase to perform additional experiments on standard benchmark tasks (see Fig. +5 on the [Project page](https://sites.google.com/view/hrl-with-timed-subgoals/home)). We report the results of these experiments in Fig. +6 on the [Project page](https://sites.google.com/view/hrl-with-timed-subgoals/home).

In the static **Pendulum** environment from [1], HiTS learns faster than HAC as already observed on the UR5Reacher environment [1]. As discussed in the Supplementary Material C.4, we hypothesize that this might be due to the lower level being able to generalize well over $\Delta t$. In contrast to this, HAC has to learn how to perform a movement faster by trial and error.

We can furthermore confirm that the performance of HiTS on **AntFourRooms** [1] can be improved by tuning the subgoal budget and optimizing the learning rates on both levels separately. However, as including these hyperparameters in the optimization also improved the performance of HAC, HiTS still learns slower than the baseline in this environment.

We moreover considered the **Ball in cup** task from the DeepMind Control Suite [2]. As the ball attached to the cup via a string is not under the direct control of the agent, the environment can be considered dynamic. Even though the task is rather simple, HiTS learns slightly faster than the strong SAC baseline. Note that HiTS succeeds despite the non-sparse interaction between cup and ball. This demonstrates that the sparse interaction assumption, which is necessary for the stationarity proof, can be relaxed in practice. While HAC can solve this simple task, it suffers from a big variance, as expected in a dynamic environment, and stagnates at a lower asymptotic performance.

In summary, HiTS can solve all of the considered tasks and outperforms HAC in three out of four environments. We therefore consider HiTS to be generally applicable to reinforcement learning problems, be they dynamic or static. Also note that the variance of HiTS on these standard benchmark tasks is lower than on the more challenging dynamic tasks we introduced. As suggested by multiple reviewers, we will include these results in the main text.

[1] A. Levy, G. Konidaris, R. Platt, and K. Saenko. Learning Multi-Level Hierarchies with
Hindsight. In International Conference on Learning Representations, 2019.

[2] Tassa et al, 2020. dm_control: Software and Tasks for Continuous Control, arXiv 2006.12983.

---

### Decision · Program_Chairs · 2021-09-27

**Decision:**

Accept (Poster)

**Comment:**

The paper tackles hierarchical learning in dynamic environments. Dynamic environments induce non-stationarity from the point of view of higher level policies, as the effect of high-level actions depends on the low-level policies that are learned concurrently. Earlier work has tried to address this by re-labelling past actions. However, in dynamic environments this is not sufficient, as background processes might continue further or less far dependent on how *fast* the lower level policy reaches its goal. The submitted manuscript investigates the use of 'timed sub goals' to address this, having sub-policies attain their goals at a specified time.

The reviewers are mixed on this submissions, with the reviewers having shifted towards a 'marginal accept' average. The main points by the reviewers are:
- The main ideas are mostly considered relevant, interesting, and original
- The execution of experiments and motivation of design decisions is technically sound. One reviewer wrote that the problem definition could be more precise.
- The paper is well written
- A main criticism with the original manuscript by a majority of the reviewers was that the chosen experiments insufficiently support the main claims made in the paper.
- - One of the main points is that the paper aims to address a common problem in hierarchical learning in all kinds of dynamic environments, whereas the experiments focus on very specific problems where precise timing is very important. While these are highly illustrative, they do not make it clear whether the method also brings benefits on problems that are dynamic but do not rely that much on very precise timing. (The paper does show results are competitive on static MDPs, with a small gap to methods specifically for that setting as expected).
- - furthermore, some concerns exists regarding the paper not being tested on common benchmark, the high variance in results, and the assumptions made.
- - The authors have since provided additional results on more standard settings, with 'HiTS' outperforming 'SAC' on 2 out of 4 environments (with results being practically the same as the baselines on a third environment, 'ball in a cup'). One concern about the additional evaluations is that it is hard to understand what factors cause the difference in performance - e.g., is it really in the timing of the subgoal, or in some other aspect of the difference between methods (e.g. reward structure for lower level policy, ...). Unfortunately, the author's reply came close to the end of the discussion period and so these aspects could not be fully discussed.

Taken together, I think that although the ideas are interesting and original, currently there are still some doubt about the thoroughness of the support of the  claims of the paper.

As an aside, one reviewer brought up an interesting point that the performance of timed subgoals might indicate that some form of communication is needed, but that other types of communication might work as well. It was mentioned as being unclear whether timed subgoal is indeed the best type of communication needed. This seems to be an interesting point for future work, although in the absence of other communication-based strategies of the problem it's not a comparison that I would have expected in the current paper.